# White matter maturation is associated with the emergence of Theory of Mind in early childhood

Charlotte Grosse Wiesmann[1,2], Jan Schreiber[1], Tania Singer[3], Nikolaus Steinbeis[3,4,*] & Angela D. Friederici[1,*]

The ability to attribute mental states to other individuals is crucial for human cognition. A milestone of this ability is reached around the age of 4, when children start understanding that others can have false beliefs about the world. The neural basis supporting this critical step is currently unknown. Here, we relate this behavioural change to the maturation of white matter structure in 3- and 4-year-old children. Tract-based spatial statistics and probabilistic tractography show that the developmental breakthrough in false belief understanding is associated with age-related changes in local white matter structure in temporoparietal regions, the precuneus and medial prefrontal cortex, and with increased dorsal white matter connectivity between temporoparietal and inferior frontal regions. These effects are independent of co-developing cognitive abilities. Our findings show that the emergence of mental state representation is related to the maturation of core belief processing regions and their connection to the prefrontal cortex.

[1] Max Planck Institute for Human Cognitive and Brain Sciences, Department of Neuropsychology, Stephanstrasse 1a, Leipzig 04103, Germany. [2] Berlin School of Mind and Brain, Humboldt-Universität zu Berlin, Unter den Linden 6, 10099 Berlin, Germany. [3] Max Planck Institute for Human Cognitive and Brain Sciences, Department of Social Neuroscience, Stephanstrasse 1a, Leipzig 04103, Germany. [4] Developmental and Educational Psychology, Institute of Psychology, Leiden University, P.O. Box 9555, 2300 RB Leiden, Netherlands. * These authors contributed equally to this work. Correspondence and requests for materials should be addressed to C.G.W. (email: wiesmann@cbs.mpg.de).

Humans have the ability to represent what other people think and believe. This implies that we are able to understand that beliefs may differ from reality and, therefore, that people can have false beliefs about the world. This ability allows us to predict how a person with a false belief about the world will act so that we can adjust our own actions accordingly. As such, Theory of Mind (ToM) constitutes a key role for complex interaction between human individuals, including behaviours such as cooperation, social communication and morality[1,2]. Understanding that others can have false beliefs is considered to be a crucial test for ToM[3]. In childhood, there is a developmental breakthrough between the ages of 3 and 4 years, when children start passing standard false belief tests[4,5]. In these tests, subjects are asked to predict how an agent with a false belief about an object, typically concerning its location, content or nature, will act[4–6]. It has been argued that the breakthrough seen in these tests reflects a fundamental change in children's understanding of other agents, and that at this age children start to build representations of others' mental states, which can thus differ from reality[6]. The behavioural emergence of a representational ToM has been studied and debated extensively. The neural mechanisms that enable this crucial step in the development of human social cognition, however, remain largely unknown. In the present study, we therefore investigated developmental changes in brain structure between the ages of 3 and 4 years that were related to the emergence of false belief understanding.

In adults, functional magnetic resonance imaging (fMRI) studies have shown that ToM tasks recruit a bilateral network including the ventromedial prefrontal cortex (vMPFC) and dorsomedial prefrontal cortex, anterior cingulate cortex, temporoparietal junction (TPJ), superior temporal sulcus (STS) and middle temporal gyrus (MTG), temporal pole, precuneus (PC) and the inferior frontal gyrus (IFG)[7,8]. Recent meta-analytic evidence suggests activation in differential networks for different types of ToM tasks, where false belief understanding specifically recruits a fronto-temporoparietal network including the TPJ, STS/MTG, PC and MPFC[7]. One drawback of such studies on false belief understanding is that they have often used tasks that have been developed for preschool-aged children in adults. It is unknown if the maturation of these brain regions implicated in false belief tasks in adults are associated with the emergence of ToM in childhood. Given the distributed network of regions involved in belief processing in adults, we hypothesized that the structural maturation of this network and its connectivity should be important for the developmental breakthrough in explicit false belief understanding between the ages of 3 and 4 years.

To date, few developmental imaging studies have been conducted and all of these were with older children aged 6–12 years, when false belief understanding is already well-established[9–12]. The only studies that approximated the age at which false belief understanding emerges were restricted to electroencephalography (EEG)[13,14]. One of these studies[13] found individual differences in resting-state alpha oscillation related to ToM performance in the right TPJ and the dorsomedial prefrontal cortex, suggesting that the maturation of these regions is relevant for the emergence of false belief understanding. Although EEG signal recorded at the scalp only allows an approximate localization of the observed effects, these findings raise the hypothesis of the relevance of the connection between these brain regions. The present neuroimaging study therefore sets out to identify which brain structures support the emergence of false belief understanding.

The consistent functional activation of a distributed network of brain regions involved in belief processing indicates that the structural connections between these regions are important for

mature ToM. The maturation of these connections should consequently be critical for the emergence of the ability in childhood. Moreover, understanding the functional role of these connections in the development of ToM might give insight into the interrelation of the different regions within the network. This can help us understand the cognitive and neural steps that lead to the developmental breakthrough in explicit false belief understanding around the age of 4 years. Furthermore, it can shed light on the functional building blocks of mature ToM and their interaction in the developed brain network. Surprisingly, despite its distributed consistent functional network, the structural network involved in ToM has been studied very little to date. Support for the relevance of structural connectivity for ToM comes from studies with patients[15,16]. A study with patients with resected gliomas along the associative white matter pathways showed that impaired dorsal connectivity from posterior temporal and parietal regions to the prefrontal cortex along the arcuate fascicle and the cingulum correlated with ToM deficits[15]. So far, however, connectivity has not been studied in the context of false belief understanding, considered as the critical test of ToM in development, and it remains an open question what role the maturation of fibre connections plays for the developmental breakthrough in ToM in early childhood.

In the present study, we therefore combined white matter measures with behavioural performance in false belief tasks. This was done by taking a developmental approach by studying children with and without false belief understanding. We hypothesized that the developmental breakthrough was related to the white matter pathways connecting those regions that are functionally involved in false belief reasoning in adults: (a) The cingulum connects the PC and MPFC, (b) the corpus callosum (CC) connects contralateral regions of the bilateral ToM network, (c) temporoparietal-prefrontal connections link the TPJ with the MPFC, ventrally via the inferior fronto-occipital fascicle (IFOF) and dorsally via the superior longitudinal fascicle or arcuate fascicle[17,18].

The maturation of different long-range white matter pathways during childhood is differentially related to cognitive development in various domains[19,20]. A method that allows the investigation of white matter is diffusion-weighted MRI (dMRI), which measures the diffusion of water molecules in the brain. This method provides parameters that reflect the structure and organization of white matter because the diffusion depends on tissue structure[21]. Fractional anisotropy (FA) is an index that describes the directionality of diffusion. It is sensitive to axonal organization and is modulated by fibre myelination and axonal growth, especially during development[22]. During childhood FA increases with age and, in the major fibre bundles, reaches its maximum around the age of 20–40 years[23]. Tractography is a method that allows the modelling of the fibre bundle pathways and the resulting number of streamlines passing a given voxel is interpreted as an index of connectivity strength[24]. In the present study, we acquired dMRI data and computed FA as well as streamline density from probabilistic tractography, which are interpreted as measures of axonal organization and connectivity strength, respectively. These measures were then set in relation to behavioural performance on two standard false belief tasks[4,5].

Children's performance on standard false belief tasks has been shown to correlate with their executive function[25] and linguistic abilities[26]. We therefore additionally assessed and controlled for a battery of executive function and language tests[27] to understand to what extent the maturation of distinct brain structures uniquely supported the emergence of false belief understanding. Moreover, in the past decade, implicit false belief tasks have been developed which showed that, already before the age of 2 years, infants display correct expectations of the actions of an agent with

a false belief in their looking behaviour[28,29]. However, whether these early implicit expectations of false-belief-related behaviour require representation of others' mental states (as argued, for example, by Baillargeon *et al.*[30]) is a matter of debate[31,32]. We therefore additionally included an implicit task of belief-related anticipatory looking[27] to control for earlier-developing precursors of explicit false belief understanding.

To summarize, the present study relates measures of white matter organization and connectivity, gained from dMRI data of 43 children aged 3 and 4 years, to their performance on standard false belief tests. At the same time, we controlled for co-developing linguistic and executive functions, as well as earlier developing implicit false-belief-related anticipation (see Methods). On the basis of functional studies with adults and older children, we hypothesized age-related changes in white matter structure near the TPJ, STS and MTG, PC, and MPFC, as well as increased white matter connectivity between these regions to correlate with children's performance in the false belief tasks.

The present study shows an association of 3- and 4-year-olds' false belief understanding with age-related changes in FA near the TPJ, MTG, PC and MPFC, as well as with the age-independent number of streamlines between temporoparietal and inferior frontal regions. These associations were independent of co-developing cognitive abilities. We conclude that the structural maturation of regions that support belief processing in adults, and their connection to the prefrontal cortex are important for the emergence of ToM in early childhood.

## Results

**Behavioural results**. The children performed two standard explicit tests of false belief understanding. In a false location task[4], children were asked about the actions and belief of a puppet who had a false belief about the location of a desired object. In a false content task[5], the children were asked about their own initial belief and an ignorant puppet's belief of the unexpected content of a familiar box (for details see Methods). The performance on an aggregate score of these tasks confirmed the well-documented breakthrough in explicit false belief understanding to occur around the age of 4 years: Three-year-old children performed significantly below chance ($M = 0.08$, s.d. $= 0.18$, $P = 0.0002$, one-sample Wilcoxon signed rank test against test-value 0.5), 4-year-olds performed marginally above chance ($M = 0.65$, s.d. $= 0.32$, $P = 0.09$), and there was a significant difference between the age groups ($P = 0.000002$, Mann–Whitney $U$-test). To control for co-developing abilities and early precursors of false belief understanding, we assessed a standardized test of children's linguistic abilities[33], a battery of executive function tests[27] and

gaze anticipation of the actions of an agent with a false belief[27] (see Methods).

**Tract-based spatial statistics analysis**. To see in which brain regions white matter structure is implicated in false belief understanding, we conducted a tract-based spatial statistics (TBSS) analysis projecting the children's FA values to a common white matter skeleton (for details see Methods)[34]. This analysis showed that higher false belief scores correlated with increased FA values in the right TPJ and posterior MTG, the white matter near the right vMPFC, right PC, left MTG, in medial portions of the left superior temporal gyrus (STG) and inferior temporal gyrus (ITG), and in the left thalamus (see Fig. 1 and Table 1; results are cluster size corrected at $P < 0.05$).

**The role of co-developing abilities**. The reported correlations of white matter structure and false belief scores remained significant when controlling for executive functions, language and belief-related anticipation as covariates in a multiple regression. Only the clusters in the vMPFC and the thalamus were no longer significant when controlling for all three executive function tests, and the cluster in the pMTG/TPJ when controlling for sentence comprehension. All other effects were specifically related to explicit false belief understanding, independently of co-developing abilities and implicit precursors (for the correlations of FA with executive functions, language and belief-related anticipation see Supplementary Methods).

**The role of age**. Next, we wanted to find out whether the effects were developmental, that is, due to age-related changes in FA, or whether they stemmed from age-independent individual differences in FA. Including age as a covariate in the regression indicated that the effects were age-related. To get a better understanding of the role of age, we performed a voxelwise commonality analysis, including FA and age as predictors for the false belief scores. To make sure age-related changes in FA specifically explained developments in explicit false belief understanding and not in other cognitive domains, we additionally controlled for all other assessed cognitive measures as covariates. Commonality analysis combines linear regressions on the predictors of interest to allow the study of the unique and shared linear contributions of intercorrelated predictors. This analysis showed that the common effect of age and FA in the reported regions significantly explained between 4 and 10% of the variance in the false belief score, over and above the unique contribution of age and of the other cognitive abilities (see Supplementary Table 1). These results indicate that the effects

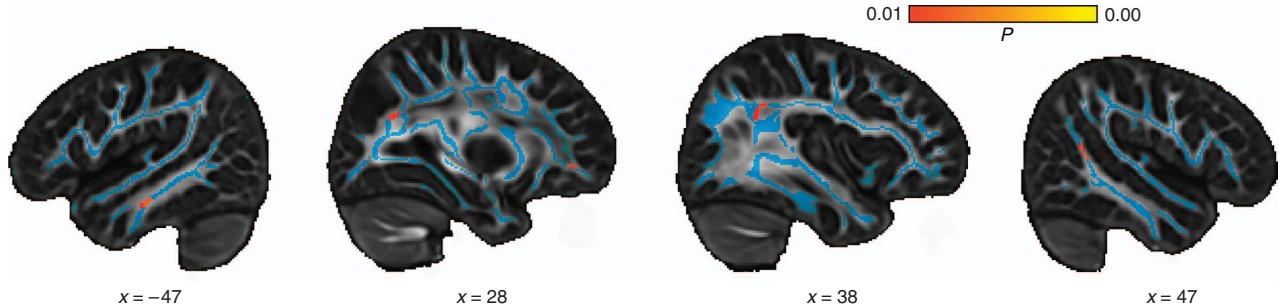

**Figure 1 | TBSS analysis.** Regions in which FA correlated with the children's false belief score ($N = 43$, voxel-level $P$ value colour coded in red-yellow, white matter skeleton in blue): white matter close to the left MTG ($x = -47$), right vMPFC and PC ($x = 28$), right TPJ ($x = 38$), and right pMTG ($x = 47$), see also Table 1. Reported coordinates are transformed to MNI (Montreal Neurological Institute) space. The results were significant at $P < 0.01$ at voxel-level and cluster size corrected at $P < 0.05$ (two-sided). The effects remained significant when co-developing abilities were controlled for.

**Table 1 | Correlations of false belief score with FA in TBSS analysis and WM tracts resulting from tractography when seeding in these regions.**

| WM in/near | MNI coordinates CoG | | | Size in voxels | P value | Correlation coefficient r | Tractography WM tracts |
|---|---|---|---|---|---|---|---|
| rTPJ | 37 | −50 | 30 | 66 | $10^{-12}$ | 0.43 | AF, ECFS |
| rpMTG/TPJ | 47 | −51 | 14 | 49 | 0.003 | 0.41 | AF, ECFS |
| lMTG | −48 | −18 | −20 | 63 | 0.0004 | 0.44 | AF, ECFS, ILF |
| lITG | −33 | −13 | −13 | 63 | 0.0002 | 0.42 | IFOF, ILF |
| lSTG | −38 | −32 | 2 | 35 | 0.01 | 0.40 | IFOF |
| rvMPFC | 28 | 46 | −4 | 31 | 0.04 | 0.41 | IFOF, CC |
| rSPL/PC | 28 | −64 | 28 | 34 | 0.01 | 0.38 | CC |
| lThalamus | −3 | −12 | 0 | 18 | 0.0004 | 0.42 | aTR |

a, anterior; AF, arcuate fascicle; CC, corpus callosum; CoG, centre of gravity; ECFS, extreme capsule fibre system; FA, fractional anisotropy; IFOF, inferior fronto-occipital fascicle; ILF, inferior longitudinal fascicle; ITG, inferior temporal gyrus; l, left; MNI, Montreal Neurological Institute; MTG, middle temporal gyrus; p, posterior; PC, precuneus; r, right; SPL, superior parietal lobule; STG, superior temporal gyrus; TBSS, tract-based spatial statistics; TPJ, temporoparietal junction; TR, thalamic radiation; vMPFC, ventromedial prefrontal cortex; WM, white matter.

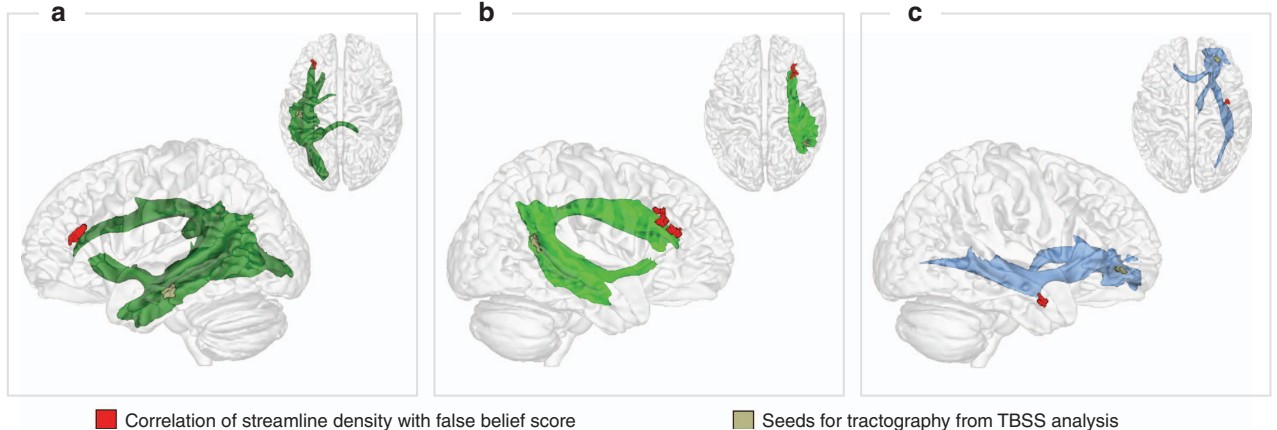

■ Correlation of streamline density with false belief score      ■ Seeds for tractography from TBSS analysis

**Figure 2 | Correlation of false belief score with streamline density.** The streamline density correlated with the children's false belief scores ($N = 43$, correlating regions in red, cluster size corrected at $P < 0.05$ (two-sided), Bonferroni corrected for the number of tracts, see also Table 2): (**a**) in the left IFG at the anterior tip of the left arcuate fascicle; (**b**) in the right IFG at the anterior tip of the right arcuate fascicle; and (**c**) in the right MTG along the IFOF. The effects were independent of age and of co-developing abilities.

**Table 2 | Correlations of false belief score with streamline density.**

| WM tract | Region of cluster | MNI coordinates CoG | | | Size in voxels | P value* | Correlation coefficient r |
|---|---|---|---|---|---|---|---|
| lAF | IFG | −30 | 37 | 17 | 61 | 0.03 | 0.46 |
| rAF | IFG | 33 | 31 | 25 | 67 | 0.002 | 0.45 |
| rIFOF | MTG | 42 | −3 | −25 | 24 | 0.002 | 0.46 |

AF, arcuate fascicle; CoG, centre of gravity; IFG, inferior frontal gyrus; IFOF, inferior fronto-occipital fascicle; l, left; MNI, Montreal Neurological Institute; MTG, middle temporal gyrus; r, right; WM, white matter.
*Adjusted according to Bonferroni for number of tracts ($n = 8$).

were indeed driven by age-related changes in white matter structure in these regions that specifically explained the development of false belief understanding.

**Connectivity analysis.** To see within which tracts the regions found in the TBSS analysis were located, probabilistic tractography was seeded in these regions. The resulting tracts are listed in Table 1 (for details see Methods). The three clusters in the right posterior MTG and TPJ and in the left MTG projected to the arcuate fascicle bilaterally, which connects posterior temporal regions with the IFG dorsally, and to the extreme capsule fibre system ventrally. Seeding in the two clusters in the more medial white matter of the left STG and ITG and in the right

vMPFC yielded streamlines along the ventral left and right IFOF. Finally, seeding in the two medial right clusters yielded an inter-hemispheric connection between the bilateral vMPFC and the bilateral PC through the CC.

Next, we wanted to specify how and where connectivity in these tracts was related to the children's false belief understanding. To this end, we correlated the streamline densities obtained from tractography with the children's individual false belief scores, while controlling for FA in the respective seed region (see Methods for details). This showed a significant correlation of the false belief score with the streamline density at the anterior tip of the left and right arcuate fascicle in the anterior IFG (Brodmann Area [BA] 45) and within the right IFOF in the MTG, (see Fig. 2 and Table 2; cluster size corrected at $P < 0.05$

and Bonferroni corrected for the number of tracts). The effect in the arcuate fascicle remained significant when controlling for age, executive functions, linguistic abilities and implicit belief-related anticipation as covariates in a multiple regression. The effect in the IFOF proved to be age-independent, but was no longer significant when all three executive function tasks or the sentence comprehension task were controlled for. An additional tractography, restricted to dorsal pathways only, confirmed that the observed effects in the IFG stemmed from streamlines of the arcuate fascicle (see Methods). Stronger dorsal connectivity from the MTG/TPJ to more anterior portions of the IFG thus specifically explained explicit false belief understanding, independently of age and of co-developing cognitive abilities.

## Discussion

A milestone of human cognitive development is reached around the age of 4 years, when children begin to understand others' false beliefs. Until now, however, no neuroimaging study had looked at the neural mechanisms that underlie this crucial step in human development. In the present study, we found that this behavioural breakthrough in ToM was associated with age-related changes in white matter in the regions involved in belief processing in fMRI studies with adults and older children. More specifically, we showed that 3- and 4-year-old children's false belief scores correlated with age-related increases in FA in the white matter around the right TPJ, left MTG, right vMPFC and right PC. These relations were independent of co-developing abilities known to correlate with ToM, that is, of children's linguistic and executive functions and of their earlier-developing implicit anticipation of belief-related actions. White matter maturation in the core ToM network thus specifically explained the emergence of explicit false belief understanding in early childhood. This complements first indications from an EEG study[13] on oscillation patterns from temporoparietal and medial prefrontal regions in relation to false belief understanding. Our results go beyond these previous findings by focusing on the maturation of brain structure, and by allowing a precise localization of the regions related to the emergence of false belief understanding. A battery of cognitive tests, moreover, allowed us to demonstrate the specificity of our effects for ToM.

White matter maturation processes reflect developmental change in the structural connections between brain areas. Probabilistic tractography with seeds in the above regions yielded a network connecting the regions functionally involved in ToM in adults: this network included the arcuate fascicle, the IFOF and the CC connecting the left and right vMPFC anteriorly and PC posteriorly. We correlated the streamline density in the pathways that resulted from tractography with the children's false belief scores to understand how the connectivity strength of these tracts was related to the developmental breakthrough in explicit false belief understanding. This approach underlined the importance of two temporoparietal-prefrontal pathways for the development of a mature representational ToM: a ventral pathway via the IFOF and a dorsal pathway via the arcuate fascicle. In the right IFOF, a correlation was found in a relatively small region in the MTG. For the arcuate fascicle, false belief understanding correlated with the streamline density at its anterior tip in IFG (BA 45) bilaterally. This correlation was age-independent and proved to be independent of language, and executive functions. This finding points to a specific role of the connection to this anterior region of the arcuate fascicle for mature false belief reasoning, independently of co-developing abilities. These results suggest that, in addition to white matter maturation in the classical belief processing regions, the extent to which temporoparietal regions are connected to anterior portions of the IFG via the arcuate

fascicle yields a mechanism for the emergence of mature human ToM.

The classical view in which false belief understanding emerges around the age of 4 years has been called into question by novel implicit tasks which show that, already before the age of 2 years, infants display different looking behaviours towards agents with a false rather than with a true belief[28,29]. Whether this sensitivity to others' belief-related behaviour is a precursor of later verbal and explicit false belief understanding has been debated intensely in the past years[30–32,35]. Against this background, we compared our findings for the standard explicit tasks of false belief understanding with an implicit task of belief-related anticipatory looking. This analysis revealed that FA only correlated with the standard explicit false belief tasks and not with the implicit task. Moreover, a commonality analysis for the explicit false belief tasks, including the implicit task as a predictor, showed that FA in the reported regions correlated with explicit false belief understanding independently of the gaze anticipation of the actions of an agent with a false belief. The association with white matter development in the reported regions thus seems to be specific for the emergence of mature explicit false belief reasoning independently of earlier-developing anticipation of belief-related behaviour. The observed independence of explicit and implicit false belief tasks is consistent with behavioural data that show that the implicit false belief tasks are passed substantially earlier and dissociate from the explicit false belief understanding in preschool-aged children[27,36,37], as well as in those with autism[38]. It is furthermore in line with an adult fMRI study which suggests that explicit and implicit false belief tasks recruit differential brain networks[39]. Future research will have to follow-up on the brain regions and connections relevant for mastering implicit false belief tasks. For this, a battery of different implicit false belief tasks should be used, including anticipatory looking as well as violation of expectation paradigms, so that the robustness and reliability of different implicit measures can be assessed. Moreover, such an approach would have to include younger children at an age when implicit abilities emerge and a developmental change in performance can be observed. Considering the difficulties in performing MRI with infants, this will remain a major challenge for future research.

In sum, the present neuroimaging results demonstrate that the emergence of a mature explicit representation of mental states is associated with age-related white matter maturation in the core ToM network (that is, the TPJ, MTG, PC and MPFC), independently of earlier implicit precursors and of other cognitive abilities. Moreover, this crucial step in human social cognition is specifically related to the extent to which temporoparietal regions connect to the anterior IFG via the arcuate fascicle.

The arcuate fascicle is known to be involved in language processing. It connects the posterior temporal region with Broca's area in the IFG, which have been described to support sentence comprehension[40,41]. Here, however, the observed correlation of the arcuate fascicle with false belief understanding was observed for streamlines connected to more anterior parts of the IFG, namely to BA 45 bordering BA 46 and was independent of the children's linguistic abilities. This more anterior part of the IFG has been observed for the processing of abstract hierarchies in non-language domains[42,43]. It has been argued that mature explicit false belief reasoning compared with lower-level ToM processes, such as anticipating others' actions based on their intentions or knowledge, requires one to build a mental representation of others' mental states[6]. Such a meta-representation requires hierarchical embedding of belief contents into the context of it being someone's mental state. Abstract hierarchy processing is therefore precisely the ability needed in addition to belief processing in order to reason about others' false

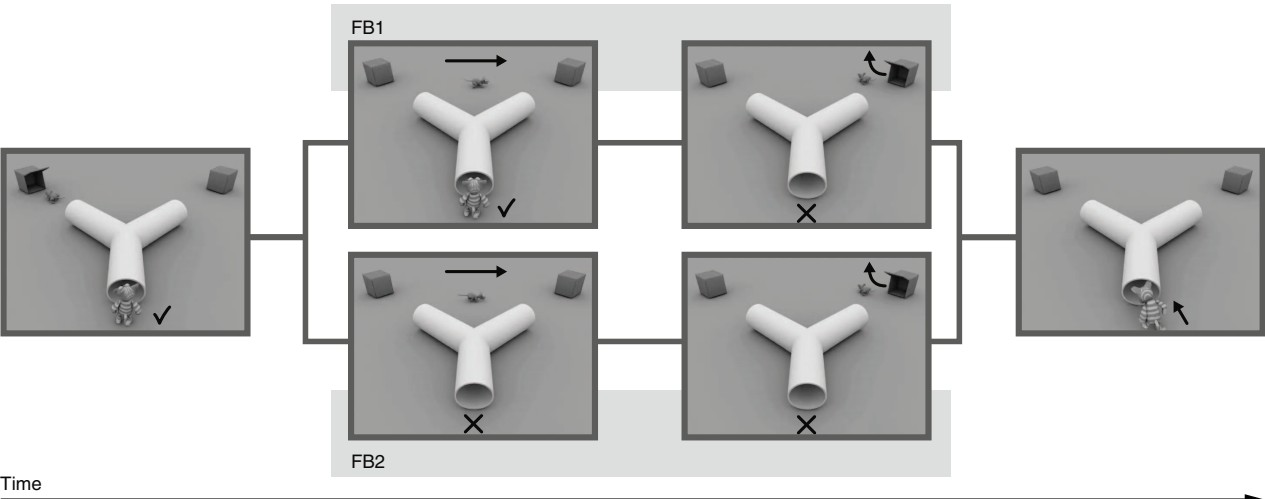

**Figure 3 | Implicit belief-related anticipatory looking task.** Selected scenes from the two false belief conditions FB1 and FB2. Arrows indicate the movement of the animals, check marks or crosses underline whether the agent animal can see what happens or not.

beliefs. We therefore suggest a mechanism in which the connection of the arcuate fascicle to more anterior portions of the IFG allows the information transfer between temporoparietal belief processing regions and hierarchical embedding of beliefs as a meta-representation in the IFG, thus enabling the developmental breakthrough in representational ToM. Future longitudinal research will have to check to what extent the relation of cognitive and brain development is causal and will need to verify that our results are truly developmental and not due to systematic individual differences between the age groups.

Taken together, our results show that white matter maturation in the brain regions associated with false belief processing (that is, the TPJ, MTG/STS, MPFC and PC) as well as the extent to which the TPJ is connected to anterior portions of the IFG via the arcuate fascicle pave the way for the emergence of the mature human ability to represent beliefs. Interestingly, in non-human primates, who have recently been shown to pass an implicit false belief task[44], but not explicit ToM abilities[46,47], the arcuate fascicle is very weak[45]. The relevance of this pathway for a full-fledged representational ToM thus points to a critical role of the arcuate fascicle for the ontogeny and phylogeny of a core aspect of human higher cognition.

## Methods

**Participants.** MRI data and behavioural data of 43 normally developing 3- and 4-year-old children (17 children aged 3–3.5 years, $M = 3.32$, s.d. $= 0.19$, 10 female; and 26 children aged 4–4.5 years, $M = 4.29$, SD $= 0.17$, 15 female) were analysed for the present study. The data of another five 3-year-olds and one 4-year-old were acquired but not analysed due to artifacts in the dMRI data set. Children were excluded if more than 10 out of 60 acquired directions in the dMRI data set were corrupted. Directions were removed due to intensity dropout caused by head motion[48] or due to artefacts detected in a visual inspection[49,50]. The sample size was based on previous developmental dMRI studies with approximately 20 children per age group, assuming a dropout rate of 10–20% due to motion artefacts. A power analysis with G*Power[51] showed that the computed correlations with $N = 43$, an effect size of $r = 0.5$, and an α-error of 5% had a power of 1-β $= 97\%$. Parental informed consent was obtained for all children in accordance with approval from the Ethics Committee at the Faculty of Medicine of the University of Leipzig.

**Cognitive assessment of the false belief score.** The children performed two standard tests of explicit false belief understanding[27]—a false location task[4] and a false content task[5]—on the same day as their MRI scans. In the false location task, each child was introduced to a mouse puppet, and they were both shown a sweet in a little bag and an empty box. The mouse then left the room, and the sweet was

moved from the bag to the box. When the mouse returned, the child was asked three probe questions about where the mouse would look for the sweet, whether she knew where it was and where she believed it was, along with a control question to make sure the child remembered the actual location of the sweet. In the false content task, the children were shown a familiar chocolate box and were asked what they believed was inside the box. Every child expected chocolates to be inside the box. They were then shown that the box actually contained pencils. The mouse puppet then entered the scene and the children were asked three probe questions: whether the mouse knew what was in the box, what she believed was in it, and what the child itself had originally believed, along with a control question on the actual content of the box. All children answered the control questions correctly in both tasks. In each of the tasks, children could obtain a total of three points, one for each of the three probe questions. The performance on the two tasks was highly intercorrelated (Spearman's $\rho(43) = 0.879$; $P = 8 \times 10^{-15}$). We therefore combined them into a total false belief score with equal weight for each of the six probe questions. This yielded a sufficiently varied and highly reliable measure (Cronbach $\alpha = 0.894$) suited to study correlations with other measures. The three questions in each of the false belief tasks could have led to carry-over effects or pragmatic pressure to give different answers to consecutive very similar questions. To exclude the possibility that such effects might have influenced our results, we replicated our analyses with a false belief score that only included the first question of each of the two false belief tasks. This scoring yielded very similar results (see Supplementary Tables 2 and 3).

**Other cognitive abilities.** To control for co-developing abilities, the children additionally performed a battery of three executive function tasks[27] as well as a standardized test of language development (SETK 3-5, Sprachentwicklungstest für drei- bis fünfjährige Kinder)[33]. Moreover, children's implicit expectations of the actions of an agent with a false belief—known to precede explicit false belief reasoning in development—were tested with an anticipatory looking false belief task[27]. These additional tests were conducted in two separate sessions before the MRI scan, all within an average period of 14.7 days (s.d. $= 6.8$).

**Executive functions.** The children were tested on a battery of three executive function tasks[27]—a Reverse Categorization task[52], a Go-NoGo task[53] and a Delay of Gratification task[54]. The tasks were chosen to tap into the children's inhibitory control, response selection and cognitive flexibility, which have been argued and shown to be particularly relevant for mastering standard false belief tests[25,27,28].

In the Reverse Categorization task, the children were asked to sort blue and red cubes of two different sizes into a big blue box and a small red box with changing rules: first matching the colours of cubes with the boxes, then the rule was reversed, next according to the cube size, and finally reversed. The percentage of correct trials in the three rounds following a rule change was encoded as dependent variable ($M = 89.9\%$, s.d. $= 11.7\%$). This measure had a very high reliability (Cronbach $\alpha = 0.899$).

In the Go-NoGo task, children were asked to perform actions a duck puppet asked them to do (for example, 'Clap your hands!'), but not to do anything the nasty crocodile asked them to. It was checked before that children understood the rules and were able to perform the movements. A d-prime value was calculated

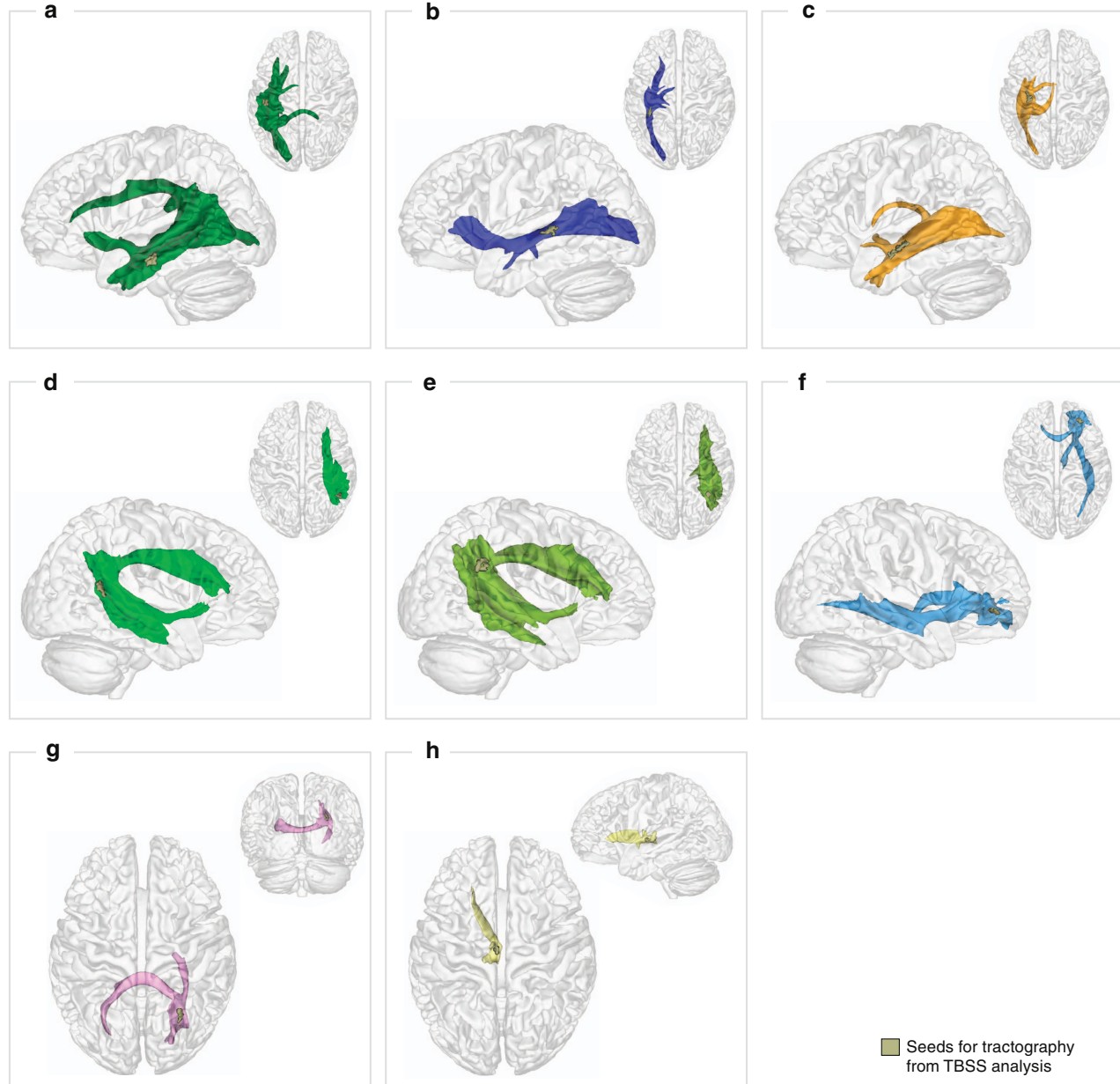

**Figure 4 | Streamline density maps resulting from probabilistic tractography seeded in the regions depicted in grey with significant correlation of FA and the false belief scores in the TBSS analysis.** Left hemisphere, (**a**) arcuate fascicle seeded in the MTG, (**b**) IFOF seeded in the STG, (**c**) ILF, IFOF and fornix seeded in WM below the ITG. Right Hemisphere: (**d**) arcuate fascicle seeded in the pMTG/TPJ, (**e**) arcuate fascicle seeded in TPJ, (**f**) IFOF seeded in vMPFC. Medial: (**g**) corpus callosum seeded in the right SPL (WM near PC), (**h**) left anterior thalamic radiation seeded in the thalamus.

Seeds for tractography from TBSS analysis

with correct NoGo-trials as hits and incorrect Go-trials as false alarms ($M = 0.886$, s.d. $= 0.172$). This task was highly reliable (Cronbach $\alpha = 0.843$).

In the Delay of Gratification task, children were seated in a small room with a small portion of their preferred sweets (gummy bears or chocolate bars) and a bell on a table in front of them. A bigger portion of the sweets was placed in a locked transparent box next to it. The experimenter told the children that she had to leave for a while, but if they waited until the experimenter came back without eating the sweets or ringing the bell to call the experimenter, they would get the big portion of sweets. Task comprehension was checked with two control questions before the children were left alone for a maximum of 5 min. The children's mean waiting time was $M = 233$ s (s.d. $= 107$ s).

We formed an aggregate executive function score for further analysis by building the mean of the $z$-scores of all three tasks (3-year-olds: M $= -0.63$, s.d. $= 1.16$; 4-year olds: $M = 0.33$, s.d. $= 0.76$; age effect: t(42) $= -3.30$, $P = 0.002$). The aggregate executive function score explained a significant amount of variance in the children's false belief scores (Spearman's $\rho = 0.520^{***}$, $P = 0.0004$), indicating that it indeed allowed us to control for the variance in false belief understanding due to the children's executive function abilities.

**Language.** As a measure of language abilities, we acquired the standardized test of language development for 3- to 5-year-old children SETK 3-5 (Sprachentwicklungstest für drei- bis fünf-jährige Kinder)[33]. The test included sentence comprehension and production, vocabulary comprehension and production, morphological rule building and phonological working memory. The mean standardized $T$-value of all subtests served as independent variable to control for children's language abilities ($M = 57.4$, s.d. $= 7.4$). This measure was significantly correlated with the false belief score (Spearman's $\rho = 0.306^*$, $P = 0.046$).

**Belief-related anticipation.** In an implicit belief-related anticipatory looking task[27], the children were presented with short film clips on a Tobii T120 eye-tracker monitor showing an animal agent observing and following a mouse through a y-shaped tunnel to one of two boxes at the two exits of the tunnel (see Fig. 3). The children were first familiarized with the fact that the animal agent would go to the box with the mouse. Then, the children were shown film clips in which the agent had a false belief about the location of the mouse, which had actually left the scene in the animal's absence. The children's anticipatory looking

was evaluated as a measure of their expectations as to where the agent would look for the mouse. There were two different false belief conditions (FB1 and FB2), respectively controlling for different non-belief-related strategies[27,29]. Every child was presented with a total of 10 familiarization (FAM) trials, 12 FB trials (six of each condition), and six true belief trials (TB1 and TB2) analogous to the FB trials, except that the agent held a true belief (TB) about the mouse's location.

Gaze data were analysed for a time of interest from the moment when the agent had disappeared in the tunnel until its reappearance in the FAM and TB conditions or until the end of the trial in the FB conditions. Two regions of interest (ROI) were defined, each covering one of the tunnel exits and the corresponding box. During the time of interest, the ROI in which the child looked first (first look), as well as the ROI with the longer gaze duration (longer look) was coded. Since both measures were highly intercorrelated ($r(43) = 0.444$, $P = 0.003$), the measures were collapsed to the mean of first and longest look for subsequent analyses[27].

The children performed significantly above chance in the FAM and TB control conditions ($M = 67.8\%$, s.d. $= 12.8\%$, $t(42) = 9.09$, $P < 0.001$), confirming that they had understood the events displayed in the film clips and showed correct anticipation when no false belief was involved. The children also performed above chance in the FB trials ($M = 53.7\%$, s.d. $= 11.2\%$, $t(42) = 2.14$, $P = 0.038$). As opposed to the standard tasks of explicit false belief understanding, there was no significant difference between age groups (3-year-olds: $M = 54.1\%$, s.d. $= 11.7\%$; 4-year olds: $M = 53.4\%$, s.d. $= 11.1\%$; $t(41) = 0.214$, $P = 0.83$). This is in line with previous literature that shows that belief-related anticipation is already achieved before the age of 2 years[28,29].

**MRI data acquisition.** The dMRI data were acquired on a Siemens 3 T TIM Trio scanner using the multiplexed echo planar imaging sequence[55,56] with a resolution of 1.9 mm isotropic (TR $= 4,000$ ms; TE $= 75.4$ ms; b-value $= 1,000$ s mm$^{-3}$; 60 directions; GRAPPA 2) reducing the scanning time to 5:32 min. A field map was acquired directly after the dMRI scan. Additionally, an anatomical scan was acquired using the MP2RAGE sequence[57] at $1.2 \times 1 \times 1$ mm resolution (TR $= 5,000$ ms; TE $= 3.24$ ms; GRAPPA 3; 5:22 min). Children were acquainted with the scanning procedure by performing a mock scan a few days before the actual scan and watched a movie of their choice on MR-compatible goggles during the scan.

**dMRI data analysis.** Before preprocessing the dMRI data, volumes affected by artefacts due to motion were removed manually, as described above. Motion itself was corrected for by rigidly aligning all volumes to the last one without diffusion weighting (b0) using flirt[58] from the FSL software package[59]. The dMRI data were then rigidly aligned to the anatomical image, which again had been rigidly aligned to the Montreal Neurological Institute standard space and was interpolated to 1 mm isotropic voxel space. Distortions were corrected using the corresponding field map. All these transformations were combined before being applied to the data to require only a single step of interpolation. The diffusion tensor was computed in every voxel within the brain volume and FA maps were derived. A common group template of the participants' FA maps was created using ANTs (Advanced Normalization Tools)[60].

**TBSS analysis.** The participants' FA maps were then correlated voxelwise with their false belief scores using TBSS[34]. TBSS projects the individual subject's maximal FA values onto a common white matter skeleton, before applying voxelwise cross-subject statistics. The skeleton was thresholded at an FA value of 0.2. The nonlinear registration was done using the group-specific template as a target image. Voxelwise statistics were then carried out with a non-parametric permutation test[61] implemented in FSL[59] with the false belief score as the dependent variable, taking into account the non-normal distribution of the data. In a next step, we controlled for the language and executive function scores, as well as for implicit belief-related anticipation by including them as covariates in the linear model[61]. In addition, we computed linear regressions where we controlled for each of the executive function tasks separately, and for all the language subtests of the SETK as separate covariates. This was done to make sure that we did not miss out on variance that was only explained by one of the subtests due to the aggregate scores. Reported clusters on the skeleton were significant at $P < 0.01$ at voxel-level and exceeded a cluster size significant at $P < 0.05$ based on local smoothness estimation on the skeleton with AFNI (3dClustSim and 3dLocalstat)[62]. In addition, a similar TBSS analysis was performed for the other cognitive domains. The correlation of FA with the executive function score and the standardized language test are reported in the Supplementary Methods. No regions of significant correlation of FA with the implicit anticipatory looking false belief task were found..

**Commonality analysis.** To get a better understanding of the role of developmental change in the effects found in the TBSS analysis, a commonality analysis[63] was computed voxelwise on the skeleton including age and FA as predictors for the false belief score. A commonality analysis allows the decomposition of the contributions of several, possibly intercorrelating, linear predictors into subcomponents explained by the unique variance of the individual predictors, as well as subcomponents explained by the shared variance of all possible

combinations of the predictors. Our commonality analysis thus allowed us to determine whether the children's false belief scores were explained by age-related increases in FA in a given voxel (common contribution of age and FA) or by age-independent individual differences in FA (unique contribution of FA), while in both cases the variance explained uniquely by age (unique contribution of age) was controlled for. In addition, the children's language, executive function and belief-related anticipation scores were included as covariates in the commonality analysis to ensure that differences in FA were specifically related to false belief understanding, independently of more general cognitive development. This analysis revealed that age-related increases in FA in the respective regions significantly explained between 4 and 10% of the variance in the false belief score, over and above the unique contribution of age and of the other cognitive abilities (details see Supplementary Table 1).

**Connectivity analysis.** To see within which tracts the significant clusters from the TBSS analysis were located, these regions were taken as seeds for probabilistic tractography with MRtrix[64] using Constrained Spherical Deconvolution as a local model[65] with the default parameters. Streamlines were started in randomly selected initialization points within the seed regions until 100,000 streamlines with a minimum length of 10 mm were obtained. The tracking followed directions with a maximum fibre orientation density (FOD) value of 0.1 and a curvature radius of at least 1 mm. The tractography was restricted to white matter. This analysis yielded the tracts shown in Fig. 4.

Streamline density maps of the individual subjects' resulting tracts were masked in the common template space by imposing that at least half the subjects have nonzero values in every voxel. This was done to ensure that correlations were not outlier-driven. The individual subjects' streamline density maps were then correlated with their false belief scores using FSL randomize[61], while controlling for the mean FA in the seed region of the tractography. This was done in order to ensure that correlations with streamline density were not driven by the correlation of the false belief score and FA found in the TBSS analysis. We then controlled for age, the language, executive function, and implicit belief-related anticipation scores by including them as covariates in the linear model. In addition, we computed linear models where we controlled for each of the executive function tasks separately, and for all the language subtests of the SETK as separate covariates. Reported clusters in the tract volumes were significant at $P < 0.001$ at voxel-level and exceeded a cluster size significant at $P < 0.05$, in addition taking into account the number of streamline density maps according to Bonferroni correction.

To confirm that the effect observed in the anterior IFG stemmed from dorsal streamlines of the arcuate fascicle, we computed an additional tractography from the seed regions in the left MTG and right TPJ, which we restricted to dorsal pathways. For this, a termination mask was defined as a plane parallel to the Sylvian fissure (see Supplementary Fig. 1).

We localized and named the clusters and tracts based on the MRI Atlas of Human White Matter[17].

**Data availability.** Data, in anonymized format (according to data protection policy in the ethics agreement), is available upon request. The publication of the script for the voxelwise commonality analysis is in preparation, and the script is available from the authors on request.

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

## Acknowledgements

This research was supported by funding from the Studienstiftung des deutschen Volkes awarded to C.G.W. and from the Jacobs Foundation to N.S. We thank Hung Nguyen Trong and Christiane Attig for their help with conducting the experiments and coding the data as well as Tomás Goucha for very helpful discussions and comments on the manuscript.

## Authors contributions

C.G.W., T.S., N.S. and A.D.F. conceived the study. C.G.W. designed the behavioural experiments with help of N.S. C.G.W. conducted the experiments and analysed the data. J.S. analysed the data and contributed analytical tools. C.G.W. wrote the manuscript. N.S., J.S. and A.D.F. edited the manuscript. N.S. and A.D.F. contributed equally.
