## [Peer Review File · Nature Communications]

Reviewers' comments:

Reviewer #1 (Remarks to the Author):

This is an interesting study and I think that the data are informative about the neurodevelopmental correlates of false belief reasoning. I especially think that the inclusion of the implicit FB measure is theoretically relevant and helpful for understanding the association between implicit and explicit measures. The relatively large sample size (for this kind of data) is also a strength.

I have some concerns and suggestions for each aspect of the manuscript.

The electrophysiological studies related to the neurodevelopmental correlates of false belief reasoning (e.g., Sabbagh et al. 2009) are much more informative than the manuscript would seem to imply (see e.g., line 72-3). Claiming that it has remained "entirely elusive" is incorrect. It is also noteworthy that the design of this study is broadly similar to that work and that the general findings from both studies are the same (i.e., that the development of the core ToM areas is associated with performance on the false belief task). I think this should be fixed both in the introduction and the discussion.

Further, motivating the study simply by saying that the neural systems in question are "unknown" took away from a more fulsome description of what this particular dependent variable might add above the prior work. Surely the connectivity analysis is the most interesting thing here but little is done to theoretically motivate why connectivity might be particularly interesting for understanding the development of explicit FB reasoning. I will add that the connection with Autism is weak, particularly given the complexity of the findings with autism and implicit/explicit theory of mind.

Methodologically, the two FB tasks were administered and scored in non-standard ways - normally these are pass fail tasks, though, there is variation in the ways the questions are asked. In the present work, three questions are asked which is problematic because they are interrelated and it's not clear how answering one should carry over to answering another. Perhaps more worrying, it is unclear as to how getting a "2" on this task would be interpretable - that is, how would one interpret the pattern of a child saying, within a particular story, that the story protagonist would look in the correct location, but then answer that the story character *believes* that the item is somewhere else. Substantially more justification for this approach needs to be included, or responses could be recoded to be in line with more standard (and arguably more valid) approaches.

Also, of the 3 EF tasks that were administered, delay of gratification is expected to have a weaker relation with FB than reverse-categorization (which should be strongest) or go/no-go. I would be more convinced about the specificity of the associations if the EF battery did not include tasks not expected to be associated with FB.

With respect to statistics, I am not an expert in the FA measures, but I would encourage serious scrutiny of the cluster-based conclusions given a recent PNAS paper suggesting that for almost all standard software packages for analyzing BOLD response, cluster statistics are associated with around 60-70% false positives (Eklund et al., 2006, doi:

10.1073/pnas.1602413113). I hope that the authors can provide compelling reasons to have confidence in their findings (beyond the fact that the methods are conventional). I also hope that someone more expert than I am with respect to the FA measure is also reviewing the work.

Finally, the conclusions are too strong (even in the title), implying a causal connection between brain development and theory of mind performance that I don't think is warranted given their cross-sectional design. There is evidence that state of age-related maturation is associated with theory of mind, but no evidence confirming that age-related maturation is a cause of theory of mind development rather than the result of wholly other processes that led to both theory of mind development and these age-related changes in the brain. Longitudinal work is necessary for this.

Reviewer #2 (Remarks to the Author):

This study on the structural correlates of ToM development is highly interesting, highly novel and on a highly important question. It should also be noted that the authors have designed an impressive study in which they use a wide range of control tasks to isolate the key Theory of Mind measures of interest. The sample is somewhat on the low size for individual differences analyses, but this sample is very difficult to acquire so I think the number of subjects is actually quite impressive. This being said is it somewhat difficult for me to assess the current work as there are some analysis issues that need to be addressed.

No measures of reliability are included, neither test-retest nor split-half. Measures can only correlate if they are reliable, thus it is crucial to include this information. In addition, I'd like to see all the correlations within and across tasks, with scatter plots, provided so we know the relationships between the measures. Finally, a hierarchical regression on structural measures, with each task entered before the explicit theory of mind score and age must be conducted. I'd also like to see such analyses in the main paper and on individual measures as well as the combined scores (which can be misleading).

Reviewer #3 (Remarks to the Author):

NCOMMS-16-12779-T

White matter maturation supports the emergence of Theory of Mind

When reviewing original research please incorporate the points below into your comments to Authors. For all other article types, such as review or progress articles, please simply provide comments to authors and editors in the boxes provided below.

A. Summary of the key results

The manuscript describes a study on the neural basis of Theory of Mind capacities. Previous

studies have aimed to do the same, but generally used inappropriate samples (i.e. too old). The current study is the first to investigate the neural basis of ToM in the adequate age range of 3-4 years old, when the primary method for ToM, the false belief task, is a sensitive and valid measure. The search for brain regions related to ToM has been ongoing, and is partly impaired by the use of methods that were developed for preschool aged children in much older children or even adults. The current attempt to explore this link in the appropriate age range should therefore be applauded. Several comments are listed below.

B. Originality and interest: if not novel, please give references

Highly original and of great interest. Novel to do this study in 3-4 year olds.

C. Data & methodology: validity of approach, quality of data, quality of presentation
Very well written, adequate approach, though I am not an expert on white matter development

D. Appropriate use of statistics and treatment of uncertainties

The 4 year olds only scored marginally above chance on the FB task. A wide literature indicates that this could be expected, as most 5 to 6 year olds pass the FB task. The poor correspondence of FB scores with age in this 3-4 years old age range limits the impact of the findings. The 4 year olds only scored marginally above chance, this might also suggest false positive scores.

E. Conclusions: robustness, validity, reliability

In the discussion on page 9, first paragraph, it remains a bit unclear to what extent these findings can be attributed to age. Age related change in white matter is the key alternative explanation to the current findings, and the authors could provide a stronger account to invalidate this argument.

F. Suggested improvements: experiments, data for possible revision

Major issues

1. Despite the many strengths of the current study, a limitation is the use of the implicit anticipatory looking task as a control task. At the moment this task is not publicly available, though a paper in press is published by the same authors. Based on the current description, the validity of this task is difficult to judge, and it is unclear whether the task is comparable to the task used in the Onishi and Baillargeon paper (2005), which indicated (preliminary) ToM skills in young infants.

2. In addition to the issue raised above, it would have been more elegant to also include younger infants to test whether white matter maturation is also associated with implicit Theory of Mind.

3. On page 7, the authors describe they controlled for executive functions, language, and belief-related anticipation as covariates in their regression. However, when controlling for age, they performed a voxelwise commonality analysis. Why was age not just included as a covariate in the main regression?

4. While the authors emphasized the importance of development, the study does not include longitudinal data, which limits the findings.

Minor issues

5. Previous evidence (Schurz et al., Carrington et al.), on brain regions related to ToM are partly based on using tasks developed for preschool aged children in adult populations. This literature should be described more critically.

6. The writing could be more condensed and at times may benefit from a check by a native speaker.

7. Were there any age related changes in implicit belief-related anticipation?

G. References: appropriate credit to previous work?

Appropriate

H. Clarity and context: lucidity of abstract/summary, appropriateness of abstract, introduction and conclusion

Very adequate

Reviewer #4 (Remarks to the Author):

A SUMMARY

The present manuscript presents a study using diffusion MRI to study the relationship between white matter maturation and behavioral changes in the ability to perform false belief tasks in young children. Children performed a battery of false belief, executive function, and language tasks and were subsequently scanned. Tract-based spatial statistics were used to correlate fractional anisotropy (FA) and false belief task performance, showing clusters in medial parietal, medial frontal, temporoparietal cortex. Probabilistic tractography from these clusters suggests they belong to the arcuate fascicle (AF) and the inferior fronto-occipital fascicle (iFOF).

B ORIGINALITY AND INTEREST

This is a nicely conducted study in a group of subjects that is not often studied. I am sure the interest will be high. The results are mainly as would be predicted based on the literature.

C/D DATA AND METHODOLOGY AND USE OF STATISTICS

I was asked to mainly comment on the methods. In general, the methods are appropriate and the study seems well conducted, although some details deserve a bit more attention. I have only minor comments.

The specificity of the claim that false belief tasks are associated with the reported clusters is based on the lack of correlation with these clusters with an aggregate measure of performance on three executive control tasks and an aggregate measure of performance on a series of tasks measuring different aspects of languages. However, the authors fail to present a justification of combining the different measures into a single score. Especially in the case of the language tasks, it might be expected that measures of comprehension,

phonological working memory, and morphological rule building rely on different neural systems, as indeed shown by work from the authors' group, and that hence they explain different variance in FA. Combining them artificially weakens the change of the language measures to explain variation in FA.

Tractography was run from the clusters reported in the TBSS analysis. From the figures, it looks like the unconstrained nature of the tracking is making it pick up some spurious correlations. For instance, the tractography shown in Figure 2a shows not only the arcuate fascicle, but it curls all the way back ventrally to the ventral pathway, reaching from the temporal lobe back into the ventral frontal cortex. Do the authors want to comment on how this affects their interpretation of the results? It seems unlikely that the seed region is connected to all the areas found in the tractography via the arcuate fascicle, more likely is that multiple fiber systems are reached by the tractography, due to the convergence of connections around the temporal-parietal junction.

The authors state that "The individual subjects' streamline density maps were then correlated with their false belief scores". What exactly is correlated? The number of streamlines? If that is the case, how much is that affected by the potential false positive I mentioned in the previous paragraph?

Can the authors please report the exact tractography settings used?

E/F CONCLUSIONS AND SUGGESTED IMPROVEMENTS

In general, although I don't doubt the results, I think the manuscript would be more informative if any TBSS results correlating with the implicit false belief task and the different language and executive tasks are also reported or if the results are directly tested against one another. For instance, the finding that the FA differences correlate with the explicit false belief tasks and not with the implicit task is an interesting result and I, for one, would like to know which system is involved in the implicit task. Similarly, if the explicit false belief at this stage of development is independent of any of the language scores, it would be interesting to know which neural system does predict behavior on those.

G REFERENCES

Appropriate

H CLARITY AND CONTEXT

Appropriate

Reviewer #1:

This is an interesting study and I think that the data are informative about the neurodevelopmental correlates of false belief reasoning. I especially think that the inclusion of the implicit FB measure is theoretically relevant and helpful for understanding the association between implicit and explicit measures. The relatively large sample size (for this kind of data) is also a strength.

We thank the reviewer for their overall positive assessment of the study and their highly constructive comments that have helped us to establish the robustness of our findings and significantly improve the manuscript.

The electrophysiological studies related to the neurodevelopmental correlates of false belief reasoning (e.g., Sabbagh et al. 2009) are much more informative than the manuscript would seem to imply (see e.g., line 72-3). Claiming that it has remained "entirely elusive" is incorrect. It is also noteworthy that the design of this study is broadly similar to that work and that the general findings from both studies are the same (i.e., that the development of the core ToM areas is associated with performance on the false belief task). I think this should be fixed both in the introduction and the discussion.

The passage has been rephrased in the Introduction and a passage added in the Discussion:

Introduction, p. 4, paragraph 2:

So far, few developmental imaging studies have been conducted and all of these were with older children aged 6 to 12 years, when false belief understanding is already well-established.¹¹⁻¹⁴ The only studies that approximated the age at which false belief understanding emerges were restricted to electroencephalography (EEG)^{15,16}. One of these studies¹⁵ found individual differences in resting-state alpha oscillation related to ToM performance in the right TPJ and the dMPFC, suggesting that the maturation of these regions is relevant for the emergence of false belief understanding. Although EEG signal recorded at the scalp only allows an approximate localization of the observed effects, these findings raise the hypothesis of the relevance of the connection between these brain regions. The present neuroimaging study therefore sets out to identify which brain structures support the emergence of false belief understanding.

Discussion, p. 10, paragraph 2:

White matter maturation in the core ToM network thus specifically explained the emergence of explicit false belief understanding in early childhood. This complements first indications from an EEG study¹⁵ on oscillation patterns from temporoparietal and medial prefrontal regions in relation to false belief understanding. Our results go beyond these previous findings by focusing on the maturation of brain structure, and by allowing a precise localization of the regions related to the emergence of false belief understanding. A battery of cognitive tests, moreover, allowed us to demonstrate the specificity of our effects for ToM.

Further, motivating the study simply by saying that the neural systems in question are "unknown" took away from a more fulsome description of what this particular dependent variable might add above the prior work. Surely the connectivity analysis is the most interesting thing here but little is done to theoretically motivate why connectivity might be particularly interesting for understanding the development of explicit FB reasoning. I will add that the connection with Autism is weak, particularly given the complexity of the findings with autism and implicit/explicit theory of mind.

We thank the Reviewer for their advice on how to improve the motivation for our study. We have now added a paragraph on the motivation to the Introduction (pasted below). The Reviewer is right that the support from autism is weak, because of the complexity of the relation to ToM. We have therefore removed the sentence on autism, and, instead, have explained further support for the relevance of structural connectivity for ToM from a patient study in more detail.

Introduction, p. 4-5:

The consistent functional activation of a distributed network of brain regions involved in belief processing indicates that the structural connections between these regions are important for mature ToM. The maturation of these connections should consequently be critical for the emergence of the ability in childhood. Moreover, understanding the functional role of these connections in the development of ToM might give insight into the interrelation of the different regions within the network. This can help us understand the cognitive and neural steps that lead to the developmental breakthrough in explicit false belief understanding around the age of 4 years. Furthermore, it can shed light on the functional building blocks of mature ToM and their interaction in the developed brain network. Surprisingly, despite its distributed consistent functional network, the structural network involved in ToM has been studied very little to date. Support for the relevance of structural connectivity for ToM comes from studies with patients.^{17,18} A study with patients with resected gliomas along the associative white matter pathways showed that impaired dorsal connectivity from posterior temporal and parietal regions to the prefrontal cortex along the arcuate fascicle and the cingulum correlated with ToM deficits.¹⁷ So far, however, connectivity has not been studied in the context of false belief understanding, considered as the critical test of ToM in development, and it remains an open question what role the maturation of fiber connections plays for the developmental breakthrough in ToM in early childhood.

Methodologically, the two FB tasks were administered and scored in non-standard ways - normally these are pass fail tasks, though, there is variation in the ways the questions are asked. In the present work, three questions are asked which is problematic because they are interrelated and it's not clear how answering one should carry over to answering another. Perhaps more worrying, it is unclear as to how getting a "2" on this task would be interpretable - that is, how would one interpret the pattern of a child saying, within a particular story, that the story protagonist would look in the correct location, but then answer that the story character *believes* that the item is somewhere else. Substantially more justification for this approach needs to be included, or responses could be recoded to be in line with more standard (and arguably more valid) approaches.

We acknowledge that false belief tasks are mostly conducted as pass or fail tasks, and have now also conducted the analysis when the two tasks are coded as pass or fail tasks. This analysis yields similar results, and shows that our findings are robust independently of the coding of the false belief measure.

For the analysis, the false belief tasks were coded as 0 or 1 based on the first and arguably most prototypical false belief question that was asked in each of the two false belief tasks (false location and false content task). The two tasks were aggregated to a common false belief score which accordingly scaled from 0 to 2 out of 2. We then conducted the TBSS analysis and streamline density analysis for this binary false belief score exactly as described in the manuscript for the original false belief score. The TBSS analysis replicated all effects shown in Figure 1 of the manuscript. Only the clusters in the thalamus and in the left inferior temporal gyrus listed in Table 1 were reduced in size and did not survive cluster-size threshold. Equally, the streamline density analysis replicated the effect in the arcuate fascicle in IFG, but the effect in the IFOF remained smaller than the cluster-size threshold.

In spite of this consistency, we believe that for the aim of the current study the aggregate measure of all three questions per task is more appropriate. Previous studies have mostly aimed at showing that a certain group of children passes the test or does not pass in a between-subjects design. In the present study, however, our aim was to correlate children's performance with their brain structure. For such a correlational design, a certain amount of variance in the measures is crucial in order to be sensitive enough to detect possible correlations. Moreover, for correlations to be meaningful it is essential to be able to assess the reliability of our score. We achieved these aims by using more than one item per task and by aggregating the items to a total score that varied from 0 to 6. Finally, and perhaps most importantly, typically, false belief questions only allow for two answer possibilities (box or bag / yes or no /...). An individual child would therefore have a 50 % chance to get a correct answer simply by guessing. Because the aim of our study was to correlate individual performance with brain measures in that same individual and not to show above chance level performance across a whole group of subjects, having only a single test question would have been problematic. We therefore had a total of 6 items (3 per task) which we aggregated to a total score. Aggregation of the items was justified due to their high intercorrelation, which confirmed that they indeed tapped into the same construct and proved the very high internal consistency and reliability of our aggregate score (Cronbach's alpha = .894).

The underlying assumption of aggregating the items is that the variance reflects the robustness of a child's ability to understand false beliefs. Thus, a child that does not at all understand false beliefs yet should answer in accordance with reality on all questions instead of with the protagonists false belief and should therefore score with 0 out of 3. In contrast, a child with a robust understanding of false beliefs should give a correct answer to all questions (i.e., 3 out of 3). Finally, children who start understanding that others' beliefs impact their behavior but whose understanding is still fragile and whose confidence levels are low, might answer correctly on some of the questions but not on others (leading to scores such as 2 out of 3) depending on the belief situation and phrasing of the test question. The variance in the aggregate false belief score thus reflects the child's understanding of false beliefs.

The figures below indeed show that the variance in the aggregate score reflected the children's answers on the first false belief question the children were asked.

(Note that all computed statistical tests were nonparametric so that the data distribution met the assumptions of the tests.)

For the reasons explained above, we decided to maintain our original coding of the false belief score in the manuscript, although reconducting the analyses with a binary score showed that our results were robust independently of the coding of the variable. We have now added justification for aggregating the questions of the false belief tasks to the manuscript:

Methods, p. 22 paragraph 1:

The performance on the two tasks was highly intercorrelated (Spearman's $\rho(43) = .879$; $p < .001$). We therefore combined them into a total false belief score with equal weight for each of the six probe questions. This yielded a sufficiently varied and highly reliable measure (Cronbach alpha = .894) suited to study correlations with other measures.

Also, of the 3 EF tasks that were administered, delay of gratification is expected to have a weaker relation with FB than reverse-categorization (which should be strongest) or go/no-go. I would be more convinced about the specificity of the associations if the EF battery did not include tasks not expected to be associated with FB.

We agree with the Reviewer that some theories on the relation of executive functions and ToM predict a stronger relation of ToM with tasks of cognitive flexibility such as reverse categorization (e.g., Cognitive Flexibility and Control Theory by Zelazo & Frye 1998) or of conflict inhibition such as go-nogo (e.g., Carlson, Moses, & Breton 2002). Other more task-related accounts, however, mainly attribute the relation of the two domains to response inhibition (e.g. Baillargeon, Scott, & He 2010), as is measured by the delay of gratification task. We therefore assessed all three types of EF tasks to make sure that we were able to control for possible variance in children's FB performance due to any of these contributors. Indeed, our data show that the children's FB score correlates with each of the three EF tasks: Spearman's $\rho = .307^*$, $p = .021$ for reverse categorization; $\rho = .379^{**}$, $p = .004$ for go-nogo; and $\rho = .267^*$, $p = .047$ for the delay of gratification. These correlations did not differ significantly from each other (Williams' t-test). Thus, all three tasks explained variance in children's FB performance that we controlled for by including them as covariates into the regression of ToM on brain structure.

Moreover, we have now also replicated our main analyses when controlling for each of the three EF tasks individually. All TBSS effects shown in Figure 1 of the manuscript and the streamline density at the anterior tip of the arcuate fascicle (Figure 2) survived when each of the EF tasks was controlled for separately. This shows that our results would have remained the same if we had only included the reverse categorization and go-nogo task as suggested by the Reviewer.

With respect to statistics, I am not an expert in the FA measures, but I would encourage serious scrutiny of the cluster-based conclusions given a recent PNAS paper suggesting that for almost all standard software packages for analyzing BOLD response, cluster statistics are associated with around 60-70% false positives (Eklund et al., 2006, doi: 10.1073/pnas.1602413113). I hope that the authors can provide compelling reasons to have confidence in their findings (beyond the fact that the methods are conventional). I also hope that someone more expert than I am with respect to the FA measure is also reviewing the work.

The paper by Eklund et al. (2016) shows that parametric methods in most software packages suffer from strongly inflated false positive rates due to invalid assumptions of these tests. In our study, however, we have used a nonparametric permutation test (FSL randomise), which according to Eklund et al. is not affected by the inflated false positive rates: “For a nominal familywise error rate of 5%, the parametric statistical methods are shown to be conservative for voxelwise inference and invalid for clusterwise inference. (...) By comparison, the

nonparametric permutation test is found to produce nominal results for voxelwise as well as clusterwise inference.” (from the abstract of Eklund et al. 2016).

Finally, the conclusions are too strong (even in the title), implying a causal connection between brain development and theory of mind performance that I don't think is warranted given their cross-sectional design. There is evidence that state of age-related maturation is associated with theory of mind, but no evidence confirming that age-related maturation is a cause of theory of mind development rather than the result of wholly other processes that led to both theory of mind development and these age-related changes in the brain. Longitudinal work is necessary for this.

We acknowledge that our cross-sectional setting does not allow concluding a causal link between brain structure and ToM. We have now rephrased the wording more carefully (by replacing “*support*” or “*enable*” by “*associated with*” or “*related to*”), have changed the title, and have included the lack of longitudinal data as a limitation to our Discussion (pasted below).

The title has been changed to: “*How do children come to understand others' beliefs? - The role of white matter maturation*”

Discussion, p. 13, paragraph 1:

Future longitudinal research will have to check to what extent the relation of cognitive and brain development is causal and will need to verify that our results are truly developmental and not due to systematic individual differences between the age groups.

Reviewer #2:

This study on the structural correlates of ToM development is highly interesting, highly novel and on a highly important question. It should also be noted that the authors have designed an impressive study in which they use a wide range of control tasks to isolate the key Theory of Mind measures of interest. The sample is somewhat on the low size for individual differences analyses, but this sample is very difficult to acquire so I think the number of subjects is actually quite impressive.

We thank the Reviewer for their positive evaluation of our study and their sound methodological inquiries that have helped us to significantly improve the foundation of our results.

No measures of reliability are included, neither test-retest nor split-half. Measures can only correlate if they are reliable, thus it is crucial to include this information.

We thank the Reviewer for urging us to show the reliability of our behavioral measures, which is indeed central to establish the validity of the reported correlations. The cognitive measures we employed are very well-established measures, that have been used many times in previous studies, and have shown to have robust correlations with each other and with other tasks (see references below for every task). Additionally, we now also computed the reliability of these measures within our sample. It was not possible to compute test-retest reliability because the children are too old by now. Instead, we computed Cronbach's alpha as a measure of internal consistency and reliability of our tests (average correlation between any two halves of the test). This proved that our tests were highly reliable:

For the false belief score, Cronbach's alpha was .894. Moreover, the high correlation between the false location and the false content task shows that these measures yield consistent results across different tests and belief situations (location versus content): Spearman's rho $r = .879$; $p < .001$. The correlation of .879 can be considered as parallel-test reliability, ensuring that the two tasks reliably tapped into the same construct. Moreover, the tasks that we conducted are very well-established and have been conducted in a multitude of studies before (see e.g. the meta-analysis by Wellman, Cross, & Watson, 2001 and many more thereafter). Previous studies and meta-analyses have shown that these tasks yield highly consistent results across different variations of the task and belief situations (Wellman et al., 2001; Wellman & Liu, 2004), as we also confirmed with the high correlation between the two tasks that we used.

For the executive function tasks reliability also proved to be very high: Cronbach alpha = .899 for the reverse categorization task and .843 for the go-nogo task. This shows that performance on these tasks was also highly consistent within an individual. For the delay of gratification test, reliability could not be computed from our sample, because we only had a single measurement per child. This test, however, is very well-established, and has been conducted in numerous studies since the 1970s. Longitudinal studies have shown the stability of performance on this task even after many years (e.g. Casey et al., 2011), highly robust correlations with other measures of self-control, personality traits, academic competence and success acquired many years later (see e.g. Duckworth & Seligman, 2016; Mischel, Shoda, & Peake, 1988; Rodriguez, Mischel, & Shoda, 1989; Shoda, Mischel, & Peake, 1990), and correlations of the task with measures of brain structure (Casey et al., 2011).

All three executive function tasks employed in our study have been conducted numerous times, and show robust correlation patterns with other executive function tasks (see e.g. Carlson & Moses, 2001; Carlson, 2005; Devine & Hughes, 2014; Rakoczy, 2010).

Finally, our language measure SETK 3-5 is a standardized test that has been validated in a norm

sample of N=116-161 3-year-olds and N=134-139 4-year-olds (depending on the subtest; Grimm, 2001). The subtests had a good to very good reliability with Cronbach's alpha ranging from .70 to .89 (average = .80) depending on the subtest and age-group (Grimm, 2001).

We have now added the reliability measures from our sample when reporting the behavioral performance of the non-standardized tasks in the Online Methods.

In addition, I'd like to see all the correlations within and across tasks, with scatter plots, provided so we know the relationships between the measures.

The correlation of the two tasks within the false belief score was very high with Spearman's $\rho = .879^{***}$; $p < .001$, as shown in the figure below:

Note that the data on these tasks were not normally distributed because of the nature of the tasks with only three items. Therefore, we made sure that all statistical tests conducted in the manuscript were non-parametric and that the data met the assumptions.

The correlation of the false belief score with the executive function score was Spearman's $\rho = .520^{***}$, $p < .001$ and with the standardized language test (SETK 3-5) was Spearman's $\rho = .306^*$, $p = .046$, as shown in the figures below:

The correlations between the tasks within the executive function scores were $\rho = .396^{**}$, $p = .003$ between the Reverse Categorization and Go-Nogo task, $\rho = .406^{**}$, $p = .002$ between the Reverse Categorization and Delay of Gratification and $\rho = .206^{(*)}$ $p = .065$ (one-tailed) between the Go-Nogo and Delay of Gratification task (figures below):

Note that the individual executive function tasks were not normally distributed due to the major developmental step in executive functions between 3 and 4 years. 4-year-olds therefore had very high performance on the measures suited also for 3-year-olds. We therefore chose a battery of three different tasks that have been validated in the tested age range (Carlson 2005; Rakoczy 2010) and aggregated these clearly intercorrelated tasks to a total score. By aggregating a number of different tasks, we obtained a sufficiently varied and approximately normally distributed score that was able to explain a substantial amount of variance in the explicit false belief tasks as is shown in the scatter plot of the executive function score and false belief above ($\rho = .520^{***}$, $p < .001$). (Kolmogorov-Smirnov test of normality: 3-yos: $p = 0.2$; 4-yos: $p = .12$). Note again that statistical tests conducted in the study were non-parametric.

Finally, the correlation between the language test and the executive function score was $r = .380^{*}$, $p = .012$ (figure below):

We acknowledge the importance of understanding the intercorrelations between the employed measures and have now reported these correlations in the Online Methods.

Finally, a hierarchical regression on structural measures, with each task entered before the explicit theory of mind score and age must be conducted. I'd also like to see such analyses in the main paper (...)

We thank the Reviewer for their suggestion to compute a hierarchical regression. The multiple regression of ToM on structural measures that we computed (with all other tasks as covariates), however, is independent of the order in which covariates are entered into the analysis (Winkler et al. 2014; Anderson & Robinson 2001; <http://fsl.fmrib.ox.ac.uk/fsl/fslwiki/Randomise/Theory>). We therefore believe that this multiple regression satisfies all the criteria for answering our question and may actually be better suited than a hierarchical regression because it remains agnostic to the order in which variables are entered. This analysis showed that ToM explained significant variance in the structural brain measures over and above the other tasks.

Moreover, we believe that the interpretation of the effect size of a multiple or hierarchical regression would be problematic in our case because of the multicollinearity between our predictors. We therefore computed a commonality analysis, which allows disentangling the unique and shared contributions of intercorrelated predictors (see e.g. Nathans, Frederick, & Nimon, 2012). In a commonality analysis, multiple regressions with all possible combinations of predictors of interest are computed. A comparison of the R^2 's then allows determining how much variance in the dependent variable is explained uniquely by one predictor, over and above all other predictors, and how much variance is explained by the shared variance of two or more intercorrelated predictors. Crucially, this procedure is independent of the order in which predictors are entered into the regression, because all possible orders of predictors are computed and taken into account. A commonality analysis including ToM and all other tasks as predictors for brain structure confirmed that the effects were independent of the other cognitive domains. The interpretation of multiple or hierarchical regression would be particularly problematic with age as a covariate, because, given that we were interested in developmental change, we were particularly interested in age-related changes in white matter structure that explained age-related improvements in ToM. Multiple (or hierarchical) regression, however, does not allow testing to what extent the age-related breakthrough in ToM was mediated by age-related changes in brain structure. Instead, this can be tested with a commonality analysis (as is widely established in the

field of developmental neuroscience; e.g. Steinbeis et al. 2012; Van den Bos et al. 2015; Fengler et al. 2016; Steinbeis et al. 2014; Van den Bos et al. 2012; Newson, & Kemps 2005; Güroğlu et al. 2011; and many more). Commonality analysis allows disentangling the effect of age-independent individual differences (unique effect of brain structure) from the effect of age-related differences in brain structure (shared effect of age and brain structure). We therefore computed a commonality analysis on ToM, including each task, age, and white matter structure as predictors. This showed that the shared variance of age and FA in the reported regions (Figure 1) explained a significant amount of variance in ToM. Thus, age-related changes in white matter structure significantly predicted ToM performance. In contrast, the relation of ToM and connectivity in the arcuate fascicle and IFOF (Figure 2) proved to be age-independent. These results were also confirmed by multiple regressions of ToM on structural measures including age (and all other tasks) as a covariate.

We acknowledge that the description and justification of our procedure had been unclear, and have now reported the results of the multiple regression and of the commonality analysis in more detail and accuracy in the Results section in the main paper.

Results, p. 8:

The role of co-developing abilities. *The reported correlations of white matter structure and false belief scores survived when controlling for executive functions, language, and belief-related anticipation as covariates in a multiple regression. (...)*

The role of age. *Next, we wanted to find out whether the effects were developmental, that is, due to age-related changes in FA, or whether they stemmed from age-independent individual differences in FA. Including age as a covariate in the regression indicated that the effects were age-related. To get a better understanding of the role of age, we performed a voxelwise commonality analysis, including FA and age as predictors for the false belief scores. To make sure age-related changes in FA specifically explained developments in explicit false belief understanding and not in other cognitive domains, we additionally controlled for all other assessed cognitive measures as covariates. Commonality analysis combines linear regressions on the predictors of interest to allow the study of the unique and shared linear contributions of intercorrelated predictors. This analysis showed that the common effect of age and FA in the reported regions significantly explained between 4% and 10% of the variance in the false belief score, over and above the unique contribution of age and of the other cognitive abilities (see Online Methods Table S1). These results indicate that the effects were indeed driven by age-related changes in white matter structure in these regions that specifically explained the development of false belief understanding.*

I'd also like to see such analyses in the main paper and on individual measures as well as the combined scores (which can be misleading).

We appreciate the Reviewer's comment that combined scores can be misleading and have now checked whether our analyses also replicate when controlling for the subtests of our combined scores individually. For this, we conducted two separate analyses to avoid an inflated number of covariates and thus overfitting or reducing the power of our analyses. In one analysis, all three EF tasks were entered as separate covariates when correlating ToM and brain measures. In a second analysis, all 5 subtests of the standardized language test SETK 3-5 were controlled for as separate covariates in the correlation.

When controlling for the three separate EF tasks, all clusters of significant correlation of ToM with FA from the TBSS analysis (Figure 1 and Table 1 of the manuscript) survived, except for the clusters in the vMPFC and the thalamus that remained subthreshold. We now report this in the Results section of the manuscript (pasted below). Equally, for the correlation of ToM with streamline density our main effects at the left and right anterior tip of the arcuate fascicle

(Figure 2 of the manuscript) remained significant when controlling for all the EF tasks individually. Only the significant cluster in the IFOF did no longer survive the cluster-size threshold. This is now also reported in the Results (pasted below).

When controlling for all individual subtests of the language test, all TBSS clusters with significant correlation of ToM with FA (Figure 1 and Table 1 of the manuscript) survived, except for the cluster in the posterior MTG, which no longer survived the cluster-size correction. The correlation of ToM with streamline density in the left and right arcuate fascicle (Figure 2) also survived controlling for all individual subtests of the SETK, whereas the effect in the IFOF did not. Controlling for the language subtests separately, showed that the FA effect in the pMTG and the streamline density effect in the IFOF disappeared when controlling for the sentence comprehension subtest of the SETK. We now report these results in the paper (pasted below).

In order to avoid excessive computation (with the voxelwise permutation test), overfitting, and reducing power through an inflated number of covariates (all subtests would comprise a total of 12 predictors), we maintained the combined scores for the analysis that included all covariates assessed in the study. This analysis yielded the results reported before. We believe that combining the different tests that assess the same construct is justified for the following reasons: For the standardized language test, we computed the mean T-value following the standardized procedure described in the test manual (SETK 3-5 Grimm 2001) based on the intercorrelation of the tasks (ranging from $r = .26^{**}$ to $.66^{***}$ with a mean correlation of $.48$). The three executive function tasks were aggregated following the same procedure as in Grosse Wiesmann et al. (2016) based on the intercorrelations of these tasks (Reverse Categorization and Go-Nogo $\rho = .396^{**}$ $p = .003$; Reverse Categorization and Delay of Gratification: $\rho = .406^{**}$ $p = .002$; and Go-Nogo and Delay of Gratification: $\rho = .206^{(*)}$ $p = .065$ (one-tailed)).

Although we believe that it is better to combine the scores for the complete model in order to reduce the number of predictors, we acknowledge that it is important to understand the effects of the individual subtests, and have therefore verified that our results survive controlling for the subtests separately as explained above, and report this in the paper.

TBSS analysis:

Results, p. 8, paragraph 1:

The reported correlations of white matter structure and false belief scores survived when controlling for executive functions, language, and belief-related anticipation as covariates in a multiple regression. Only the clusters in the vMPFC and the thalamus were no longer significant when controlling for all three executive function tests, and the cluster in the pMTG/TPJ when controlling for sentence comprehension. All other effects were specifically related to explicit false belief understanding, independently of co-developing abilities

Online methods, p. 26, paragraph 2:

Additionally, we computed linear regressions where we controlled for each of the executive function tasks separately, and for all the language subtests of the SETK as separate covariates. This was done to make sure that we did not miss out on variance that was only explained by one of the subtests due to the aggregate scores.

Streamline Density analysis:

Results, p. 9, paragraph 2:

The effect in the arcuate fascicle survived when controlling for age, executive functions, linguistic abilities, and implicit belief-related anticipation as covariates in a multiple regression. The effect in the IFOF proved to be age-independent, but was no longer significant when all three executive function tasks or the sentence comprehension task were controlled for.

(...) Stronger dorsal connectivity from the MTG/TPJ to more anterior portions of the IFG thus specifically explained explicit false belief understanding, independently of age and of co-developing cognitive abilities.

Online methods, p. 28, paragraph 1:

We then controlled for age, the language, executive function, and implicit belief-related anticipation scores by including them as covariates in the linear model. Additionally, we computed linear models where we controlled for each of the executive function tasks separately, and for all the language subtests of the SETK as separate covariates.

Reviewer #3:

The manuscript describes a study on the neural basis of Theory of Mind capacities. Previous studies have aimed to do the same, but generally used inappropriate samples (i.e. too old). The current study is the first to investigate the neural basis of ToM in the adequate age range of 3-4 years old, when the primary method for ToM, the false belief task, is a sensitive and valid measure. The search for brain regions related to ToM has been ongoing, and is partly impaired by the use of methods that were developed for preschool aged children in much older children or even adults. The current attempt to explore this link in the appropriate age range should therefore be applauded.

Highly original and of great interest. Novel to do this study in 3-4 year olds.

Very well written, adequate approach, though I am not an expert on white matter development.

We thank the Reviewer for their positive evaluation of our study and for their highly valuable comments that have strongly contributed to improving the soundness of our findings and their discussion in the manuscript.

D. Appropriate use of statistics and treatment of uncertainties

The 4 year olds only scored marginally above chance on the FB task. A wide literature indicates that this could be expected, as most 5 to 6 year olds pass the FB task. The poor correspondence of FB scores with age in this 3-4 years old age range limits the impact of the findings. The 4 year olds only scored marginally above chance, this might also suggest false positive scores.

We concur with the Reviewer that it could be expected from the literature that some 4-year-olds pass the standard FB tasks while other 4-year-olds still fail on the test. This is reflected by the fact that performance is marginally above chance across the whole group of 4-year-olds. However, the figure below shows that only very few children performed around chance level but that most children either passed or failed the task. This is also reflected by the very high reliability and internal consistency of the false belief score (Cronbach alpha = .894). The interindividual variance is exactly what we were aiming at in order to be able to correlate task performance with interindividual differences in brain structure.

Moreover, note that the standard false belief tasks do not develop from chance level to above chance performance, but children first systematically indicate the wrong answer (the actual location of the object), that is, show below chance performance as the 3-year-olds do, and then start shifting to the correct answer around the age of 4 years. In fact, there was a strong relation between FB and age (correlation: Spearman's rho = .674***; $p < .001$).

The very high reliability of the false belief score speaks for robust correlations with the measure and against an increased risk of false positive results.

E. Conclusions: robustness, validity, reliability

In the discussion on page 9, first paragraph, it remains a bit unclear to what extent these findings can be attributed to age. Age related change in white matter is the key alternative explanation to the current findings, and the authors could provide a stronger account to invalidate this argument.

We thank the Reviewer for pointing out that we have not been clear enough on the role of age for our findings. Given that we were interested in developmental change, we were particularly interested in age-related changes in white matter structure that were able to explain the age-related breakthrough in ToM. Indeed, we found that age-related changes in FA in the regions shown in Figure 1 significantly explained FB performance. On the other hand, the connectivity effects in the arcuate fascicle shown in Figure 2 were age-independent (as was shown by controlling for age in a linear regression).

We demonstrated that age-related changes in FA mediated the age-related improvement in ToM with a commonality analysis, as is widely established in the field of developmental neuroscience (e.g. Steinbeis et al. 2012; Van den Bos et al. 2015; Fengler et al. 2016; Steinbeis et al. 2014; Van den Bos et al. 2012; Newson, & Kemps 2005; Güroğlu et al. 2011). These studies agree on that the declared aim of a developmental study is to identify age-related changes in the brain that explain developmental changes in behavior, as we have shown for our TBSS FA effects shown in Figure 1.

We have now extended the explanation of our commonality analysis in the Results section, and have tried to be clearer on which of our results stem from age-related development in brain structure and which of the effects are due to age-independent individual difference in brain structure.

Results, p. 8, paragraph 2:

The role of age. Next, we wanted to find out whether the effects were developmental, that is, due to age-related changes in FA, or whether they stemmed from age-independent individual differences in FA. Including age as a covariate in the regression indicated that the effects were age-related. To get a better understanding of the role of age, we performed a voxelwise commonality analysis, including FA and age as predictors for the false belief scores. To make sure age-related changes in FA specifically explained developments in explicit false belief understanding and not in other cognitive domains, we additionally controlled for all other assessed cognitive measures as covariates. Commonality analysis combines linear regressions on the predictors of interest to allow the study of the unique and shared linear contributions of intercorrelated predictors. This analysis showed that the common effect of age and FA in the reported regions significantly explained between 4% and 10% of the variance in the false belief score, over and above the unique contribution of age and of the other cognitive abilities (see Online Methods Table S1). These results indicate that the effects were indeed driven by age-related changes in white matter structure in these regions that specifically explained the

development of false belief understanding.

Discussion, p.10, paragraph 2:

In the present study, we found that this behavioral breakthrough in ToM was associated with age-related changes in white matter in the regions involved in belief processing in fMRI studies with adults and older children. More specifically, we showed that 3- and 4-year-old children's false belief scores correlated with age-related increases in FA in the white matter around the right TPJ, left MTG, right vMPFC, and right PC.

F. Suggested improvements: experiments, data for possible revision

Major issues

1. Despite the many strengths of the current study, a limitation is the use of the implicit anticipatory looking task as a control task. At the moment this task is not publicly available, though a paper in press is published by the same authors. Based on the current description, the validity of this task is difficult to judge, and it is unclear whether the task is comparable to the task used in the Onishi and Baillargeon paper (2005), which indicated (preliminary) ToM skills in young infants.

The implicit anticipatory looking false belief task employed in the present study adopts the logic and set up of the previously used anticipatory looking tasks that have been employed to test for implicit false belief understanding in infants and preschoolers (Clements & Perner, 1994; He, Bolz, & Baillargeon, 2012; Low, 2010; Meristo et al., 2012; Senju, Southgate, Snape, Leonard, & Csibra, 2011; Senju, Southgate, White, & Frith, 2009; Southgate, Senju, & Csibra, 2007; Surian & Geraci, 2012; Thoermer, Sodian, Vuori, Perst, & Kristen, 2012). The sequence of events as well as the setting of the scene exactly follow previously conducted tasks (Southgate, Senju, & Csibra, 2007; Senju, Southgate, Snape, Leonard, & Csibra, 2011; Senju, Southgate, White, & Frith, 2009; Meristo et al., 2012; Surian & Geraci, 2012; Krupenye, Kano, Hirata, Call, & Tomasello 2016). Note that we have conducted an anticipatory looking task instead of a violation of expectation task as in Onishi and Baillargeon (2005) because anticipatory looking is considered to be more robust and easier to interpret (see e.g. Southgate, Senju, & Csibra, 2007; Oakes 2012), and violation of expectation paradigms are more appropriate for infants than for the age range we studied. The logical, structural and visual similarities makes it highly comparable to the literature.

The article that describes and establishes our task in detail is now available at <http://onlinelibrary.wiley.com/doi/10.1111/desc.12445/full> (Grosse Wiesmann, Friederici, Singer, & Steinbeis, 2016). In this article, extensive analyses have been conducted in the Supporting Information as a validation of the task and to show the stability of performance on the task over trials. This also holds for the present sample: A repeated measures ANOVA across the false belief trials showed that the children's performance was stable over time ($F(11,264) = .189$; $p = .998$).

Nonetheless, even given the strong similarity of our task with previous anticipatory looking false belief tasks and its stability over trials, we concur with the Reviewer that the robustness and validity of implicit false belief tasks in general still need to be established more firmly. This, however, is a limitation that applies to all implicit false belief tasks employed so far. Some support for the predictive value and validity of the implicit false belief tasks comes from a longitudinal study that showed that anticipatory looking in a task similar to the one we used in the present study predicted children's later performance on standard explicit false belief tasks (Thoermer et al. 2012). Future research will need to follow up on such analyses to better

establish the validity and robustness of implicit false belief tasks. We have added these general limitations of implicit false belief tasks to our discussion and the Methods section (pasted below). We feel, however, that addressing this question is beyond the scope of the present study, which focused on the brain basis of the development of mature explicit false belief understanding. Studying the brain basis of earlier-developing implicit false belief tasks would have required testing infants and toddlers with MRI, which presents a major challenge due to the difficulty to have this age group participate and keep still in the scanner.

Discussion, p. 12 paragraph 1:

Future research will have to follow up on the brain regions and connections relevant for mastering implicit false belief tasks. For this, a battery of different implicit false belief tasks should be used, including anticipatory looking as well as violation of expectation paradigms so that the robustness and reliability of different implicit measures can be assessed. Moreover, such an approach would have to include younger children at an age when implicit abilities emerge and a developmental change in performance can be observed. Considering the difficulties in performing MRI with infants, this will remain a major challenge for future research.

2. In addition to the issue raised above, it would have been more elegant to also include younger infants to test whether white matter maturation is also associated with implicit Theory of Mind.

We agree with the Reviewer that it would have been interesting to include younger infants, especially for testing implicit false belief abilities. As mentioned above, however, testing toddlers with MRI is currently not feasible with the required data quality unless children would have been sedated. We have included this limitation to our Discussion as pasted above. However, we believe that our age range was appropriate for our main research question, which was to study the neural basis of the developmental step in explicit false belief understanding between the ages of 3 and 4 years, and therefore does not impact our main findings.

3. On page 7, the authors describe they controlled for executive functions, language, and belief-related anticipation as covariates in their regression. However, when controlling for age, they performed a voxelwise commonality analysis. Why was age not just included as a covariate in the main regression?

We thank the Reviewer for pointing out that our approach was unclear at this point. We have now added the results of the regression when including age as a covariate. This showed that the TBSS effects (Figure 1) were age-related as reported in the paper. The commonality analysis was performed to get a better understanding of these age-related effects, that is, to test whether age-related changes in FA mediated the age-related improvements in ToM, as explained above. We have now explained our approach in more detail in the Results Section, and have emphasized more clearly that the TBSS effects were age-related in the Discussion as pasted above (see answer to point E).

4. While the authors emphasized the importance of development, the study does not include longitudinal data, which limits the findings.

We acknowledge that the study would have benefited from longitudinal data, and have added this limitation to the Discussion (pasted below). Especially causal claims are not possible with our cross-sectional setting, and we have taken care to rephrase wordings that suggested such a causality. Future research should try to take the challenge to check the direction of effect of the relation in a longitudinal setting and verify that our results are truly developmental and not due

to systematic individual differences related to age or between the two age groups. However, the difficulties to assess MRI data in preschoolers and the very high drop-out rates of children who refuse to participate in the MRI limit the possibility of longitudinal testing in the age-group of interest. Nevertheless, we are confident that our age-related effects are due to developmental changes rather than cohort effects because of the relatively small age range, the homogeneous recruiting procedure and because the two groups had equal gender distribution, IQ and normed language scores.

Discussion, p. 13, end of paragraph 1:

Future longitudinal research will have to check to what extent the relation of cognitive and brain development is causal and will need to verify that our results are truly developmental and not due to systematic individual differences between the age groups.

Minor issues

5. Previous evidence (Schurz et al., Carrington et al.), on brain regions related to ToM are partly based on using tasks developed for preschool aged children in adult populations. This literature should be described more critically.

We have now added this criticism to the Introduction, p. 3-4:

Recent meta-analytic evidence suggests activation in differential networks for different types of ToM tasks, where false belief understanding specifically recruits a fronto-temporoparietal network including the TPJ, STS/MTG, PC, and MPFC.⁹ One drawback of studies on false belief understanding is that they have often used tasks that have been developed for preschool-aged children in adults. Moreover, the brain regions that support the emergence of false belief understanding in development are currently unknown.

6. The writing could be more condensed and at times may benefit from a check by a native speaker.

The manuscript has now been proofread by a native speaker.

7. Were there any age related changes in implicit belief-related anticipation?

There was no significant age difference between the age groups nor correlation with age in the implicit anticipatory looking task ($r = -.037$, $p = .813$). This was in accordance with our expectations from the literature that shows that already infants around in their second year of life start passing these tests and that even adults do not perform significantly better on the tasks than infants (e.g. Southgate et al. 2007; Senju et al. 2009). We report this in the Online Methods:

Online Methods, p. 25, paragraph 1:

As opposed to the standard tasks of explicit false belief understanding, there was no significant difference between age groups (3-year-olds: $M = 54.1\%$, $SD = 11.7\%$; 4-year-olds: $M = 53.4\%$, $SD = 11.1\%$; $t(41) = .214$, $p = .83$), in line with previous literature that shows that belief-related anticipation is already achieved before the age of 2 years.^{30,31}

Reviewer #4:

This is a nicely conducted study in a group of subjects that is not often studied. I am sure the interest will be high. The results are mainly as would be predicted based on the literature.

I was asked to mainly comment on the methods. In general, the methods are appropriate and the study seems well conducted, although some details deserve a bit more attention. I have only minor comments.

We thank the Reviewer for their positive assessment of our study and their valuable comments that have significantly helped us to improve the methodological foundation of the study and informative value of the manuscript.

The specificity of the claim that false belief tasks are associated with the reported clusters is based on the lack of correlation with these clusters with an aggregate measure of performance on three executive control tasks and an aggregate measure of performance on a series of tasks measuring different aspects of languages. However, the authors fail to present a justification of combining the different measures into a single score. Especially in the case of the language tasks, it might be expected that measures of comprehension, phonological working memory, and morphological rule building rely on different neural systems, as indeed shown by work from the authors' group, and that hence they explain different variance in FA. Combining them artificially weakens the change of the language measures to explain variation in FA.

We acknowledge that combined scores can be misleading and have now checked that our analyses also replicate when controlling for the individual subtests that comprised the aggregate scores as is reported in detail below and has been added to the manuscript. This was done by controlling for all five individual language subtests in one regression, and for all three executive function (EF) tests in a second regression. For the complete linear model with all predictors and for the commonality analysis we maintained the aggregate language and EF scores to avoid excessive computation, overfitting, and reducing power through an inflated number of covariates. (Taken together, all subtests would have comprised a total of 12 predictors, which would have implied computing $(2^{12} - 1) = 4095$ regressions in every voxel of the brain in the commonality analysis with a computationally intensive permutation test.)

The aggregate language T-value was formed following the standardized procedure described in the test manual (SETK 3-5 Grimm 2001) based on the considerable intercorrelations of the subtests (as verified in a big norm sample of $N = 110-156$ for each age group) that ranged from $r = .26^{**}$ to $r = .66^{***}$ with a mean correlation of $r = .48$. The three executive function tasks were aggregated following the same procedure as in Grosse Wiesmann et al. (2016) based on the intercorrelations these tasks (Reverse Categorization and Go-Nogo $\rho = .396^{**}$ $p = .003$; Reverse Categorization and Delay of Gratification: $\rho = .406^{**}$ $p = .002$; and Go-Nogo and Delay of Gratification: $\rho = .206^{(*)}$ $p = .065$ (one-tailed)).

Nevertheless, we agree with the Reviewer that the variance of individual subtests might be obscured by the aggregate scores, and have therefore now computed additional linear models where the subtests of the SETK (in one model) and the three EF tasks (in a second model) were included as separate covariates. Moreover, we agree that the different language measures might be expected to make different contributions to ToM with the strongest contribution expected from the syntax measure (Astington & Jenkins, 1999; de Villiers & Pyers, 2002; Grosse Wiesmann, Friederici, Singer, & Steinbeis, 2016; Milligan, Astington, & Dack, 2014). We therefore additionally checked our results when controlling for each of the subtests in separate regressions.

When controlling for all individual subtests of the language test, all TBSS clusters with significant correlation of ToM and FA (Figure 1 and Table 1 of the manuscript) survived, except for the cluster in the posterior MTG, which no longer survived the cluster-size correction. We now report this in the Results section of the manuscript (pasted below). The correlation of ToM with streamline density in the left and right arcuate fascicle (Figure 2) also survived controlling for all individual subtests of the SETK, whereas the effect in the IFOF did not. Controlling for the language subtests separately, showed that the FA effect in the pMTG and the streamline density effect in the IFOF disappeared when controlling for the syntax subtest of the SETK (subtest “sentence comprehension”). We now report these results in the paper (pasted below).

When controlling for the three separate EF tasks, all clusters of significant correlation of ToM with FA from the TBSS analysis (Figure 1 and Table 1) survived, except for the clusters in the vMPFC and the thalamus that remained subthreshold. Equally, for the correlation of ToM with streamline density our main effects at the left and right anterior tip of the arcuate fascicle (Figure 2) remained significant when controlling for all the EF tasks individually. Only the significant cluster in the IFOF did no longer survive the cluster-size threshold. This is now also reported in the Results section:

TBSS analysis:

Results, p. 8, paragraph 1:

The reported correlations of white matter structure and false belief scores survived when controlling for executive functions, language, and belief-related anticipation as covariates in a multiple regression. Only the clusters in the vMPFC and the thalamus were no longer significant when controlling for all three executive function tests, and the cluster in the pMTG/TPJ when controlling for sentence comprehension. All other effects were specifically related to explicit false belief understanding, independently of co-developing abilities and implicit precursors (for the correlations of FA with executive functions, language, and belief-related anticipation see Online Methods).

Online methods, p. 26, paragraph 2:

Additionally, we computed linear regressions where we controlled for each of the executive function tasks separately, and for all the language subtests of the SETK as separate covariates. This was done to make sure that we did not miss out on variance that was only explained by one of the subtests due to the aggregate scores.

Streamline Density analysis:

Results, p. 9, paragraph 2:

The effect in the arcuate fascicle survived when controlling for age, executive functions, linguistic abilities, and implicit belief-related anticipation as covariates in a multiple regression. The effect in the IFOF proved to be age-independent, but was no longer significant when all three executive function tasks or the sentence comprehension task were controlled for. (...) Stronger dorsal connectivity from the MTG/TPJ to more anterior portions of the IFG thus specifically explained explicit false belief understanding, independently of age and of co-developing cognitive abilities.

Online methods, p. 28, paragraph 1:

We then controlled for age, the language, executive function, and implicit belief-related anticipation scores by including them as covariates in the linear model. Additionally, we computed linear models where we controlled for each of the executive function tasks separately, and for all the language subtests of the SETK as separate covariates.

Tractography was run from the clusters reported in the TBSS analysis. From the figures, it looks like the unconstrained nature of the tracking is making it pick up some spurious

correlations. For instance, the tractography shown in Figure 2a shows not only the arcuate fascicle, but it curls all the way back ventrally to the ventral pathway, reaching from the temporal lobe back into the ventral frontal cortex. Do the authors want to comment on how this affects their interpretation of the results? It seems unlikely that the seed region is connected to all the areas found in the tractography via the arcuate fascicle, more likely is that multiple fiber systems are reached by the tractography, due to the convergence of connections around the temporal-parietal junction.

The authors state that "The individual subjects' streamline density maps were then correlated with their false belief scores". What exactly is correlated? The number of streamlines? If that is the case, how much is that affected by the potential false positive I mentioned in the previous paragraph?

We thank the Reviewer for pointing out that the temporal and temporo-parietal seed regions indeed connected to dorsal as well as ventral fiber pathways. We have now corrected Table 1 and added the inferior longitudinal fascicle, and extreme capsule fiber system to the white matter tracts that result from the tractography. In order to understand which of these pathways are relevant for the development of false belief understanding, we correlated the number of streamlines in a given voxel with the children's false belief scores. This yielded the effects shown in Figure 2 in dorsal anterior IFG (BA 45). In order to make sure that the significant effect that we reported indeed stemmed from dorsal streamlines of the arcuate fascicle and not from ventral streamlines, we recomputed the tractography imposing that streamlines ended at the Sylvian Fissure. For this, we defined a termination mask as a plane parallel to the Sylvian Fissure descending from Heschl's Gyrus as shown in the Figure below:

The effect in the anterior IFG remained as before with the new restricted tractography, as shown in the Figure below. This confirms that the reported effect indeed stems from the dorsal streamlines of the arcuate fascicle.

We now also report this confirmation of our effect in the Results on p. 8 paragraph 3:
An additional tractography, restricted to dorsal pathways only, confirmed that the observed effects in the IFG stemmed from streamlines of the arcuate fascicle (Online Methods).

Online Methods, p. 28, paragraph 2:

To confirm that the effect observed in anterior IFG stemmed from dorsal streamlines of the arcuate fascicle, we computed an additional tractography from the seed regions in the left MTG and right TPJ, which we restricted to dorsal pathways. For this a termination mask was defined as a plane parallel to the sylvian fissure as shown in Figure S3.

p. 35:

Figure S3: A plane parallel to the sylvian fissure was defined as a termination mask for an additional restricted probabilistic tractography from the seed regions in left MTG and right TPJ. This confirmed that the observed significant effect in the IFG stemmed from dorsal and not ventral streamlines.

Can the authors please report the exact tractography settings used?

The tractography settings have been added to the Online Methods, p. 27, paragraph 2:
(...) these regions were taken as seeds for probabilistic tractography with MRtrix⁶⁵ using Constrained Spherical Deconvolution (CSD) as a local model⁶⁶ with the default parameters. Streamlines were started in randomly selected initialization points within the seed regions until 100 000 streamlines with a minimum length of 10 mm were obtained. The tracking followed directions with a maximum FOD value of 0.1 and a curvature radius of at least 1 mm. The tractography was restricted to white matter.

In general, although I don't doubt the results, I think the manuscript would be more informative if any TBSS results correlating with the implicit false belief task and the different language and executive tasks are also reported or if the results are directly tested against one another. For instance, the finding that the FA differences correlate with the explicit false belief tasks and not with the implicit task is an interesting result and I, for one, would like to know which system is involved in the implicit task. Similarly, if the explicit false belief at this stage of development is independent of any of the language scores, it would be interesting to know which neural system does predict behavior on those.

Following the Reviewer's suggestion, we have now also computed TBSS analyses for the correlation of FA with implicit false belief, language, and executive functions (EF). Given that the focus of the study was to explain how white matter relates to ToM development, however, we feel that reporting the effects of all 9 individual subtests would exceed the scope of the manuscript. We have therefore restricted ourselves to reporting the correlation with the aggregate language and EF scores in the Online Methods (pasted below). Justification for the aggregation of the different subtests has been provided above.

Online Methods, p. 28 paragraph 4 ff.:

D. TBSS analyses for other cognitive domains.

Similar to the TBSS analysis for the explicit false belief score, a TBSS analysis was performed to assess the correlation of FA with the executive function score, the standardized language test, and the implicit belief-related anticipatory looking task. Additionally, as for the false belief score, a voxelwise commonality analysis was performed on the skeleton including age and FA as predictors for the dependent variable of interest to test whether the effects were age-related.

Executive functions. A TBSS analysis revealed a significant correlation of the executive function score with FA in the left lingual gyrus (O5) (MNI coordinates center of gravity (CoG): $x = -22, y = -65, z = 0$), which has been shown to be involved in attentional modulation towards

different visual features of objects⁶⁷ and in categorization⁶⁸. A second effect was found with FA in the hand area of the right postcentral gyrus (S1) (MNI coordinates CoG: $x = 37$, $y = -29$, $z = 51$). The TBSS commonality analysis showed that the effects both stemmed from age-related changes in FA: The shared contribution of age and FA in O5 explained 8% and of age and FA in S1 explained 9.6% of variance in the false belief score.

Language. For language, a TBSS analysis yielded one effect in the white matter bordering the right IFG (MNI coordinates CoG: $x = 34$, $y = 25$, $z = 31$) which proved to be age-independent in the commonality analysis (unique contribution of FA in right IFG: 21.4%; contributions including age: not significant). Two further clusters made no significant contribution in the commonality analysis. Broca's area in the IFG is known to support language processing bilaterally in early childhood with increasing left lateralization in the course of development.⁶⁹

Belief-related anticipation. A TBSS analysis of the correlation of FA with the percent correct anticipatory looking in the FB trials of the implicit belief-related anticipatory looking task revealed no significant correlation of FA and implicit belief-related anticipation. Future research should follow up on the brain regions and connections relevant for mastering implicit false belief tasks, possibly using a battery of different tasks including anticipatory looking as well as violation of expectation paradigms. Such an approach should also study younger infants at an age when the ability emerges and an age-difference in performance can be observed.

References:

- Anderson MJ, Robinson J. Permutation Tests for Linear Models. *Aust New Zeal J Stat Stat.* 2001;43(1):75-88.
- Astington, J. W., & Jenkins, J. M. (1999). A longitudinal study of the relation between language and theory-of-mind development. *Developmental Psychology*, 35(5), 1311–20.
- Carlson, S. M. (2005). Developmentally Sensitive Measures of Executive Function in Preschool Children. *Developmental Neuropsychology*, 28(2), 595–616. <http://doi.org/10.1207/s15326942dn2802>
- Carlson, S. M., & Moses, L. J. (2001). Individual Differences in Inhibitory Control and Children's Theory of Mind. *Child Development*, 72(4), 1032–1053. <http://doi.org/10.1111/1467-8624.00333>
- Casey, B. J., Somerville, L. H., Gotlib, I. H., Ayduk, O., Franklin, N. T., Askren, M. K., & Jonides, J. (2011). Behavioral and neural correlates of delay of gratification 40 years later, 1–6. <http://doi.org/10.1073/pnas.1108561108>
- Clements, W. A., & Perner, J. (1994). Implicit Understanding of Belief. *Cognitive Development*, 9, 377–395.
- de Villiers, J. G., & Pyers, J. E. (2002). Complements to cognition: a longitudinal study of the relationship between complex syntax and false-belief-understanding. *Cognitive Development*, 17, 1037–1060.
- Devine, R. T., & Hughes, C. (2014). Relations Between False Belief Understanding and Executive Function in Early Childhood: A Meta-Analysis. *Child Development*, 85(5), 1777–1794. <http://doi.org/10.1111/cdev.12237>
- Duckworth, A. L., & Seligman, M. E. P. (2016). Self-Discipline Outdoes IQ in Predicting Academic Performance of Adolescents, 939–944.
- Fengler, A., Meyer, L., & Friederici, A. D. (2016). *NeuroImage*, 129, 268–278.
- Grimm, H. (2001). *Sprachentwicklungstest für drei-bis fünfjährige Kinder: SETK 3-5; Diagnose von Sprachverarbeitungs-fähigkeiten und auditiven Gedächtnisleistungen*. Göttingen: Hogrefe, Verlag für Psychologie.
- Grosse Wiesmann, C., Friederici, A. D., Singer, T., & Steinbeis, N. (2016). Implicit and Explicit False Belief Development in Preschool Children. *Developmental Science*.
- Güroğlu, B., van den Bos, W., van Dijk, E., Rombouts, S. A., & Crone, E. A. (2011). *NeuroImage*, 57(2), 634-641.
- He, Z., Bolz, M., & Baillargeon, R. (2012). 2.5-year-olds succeed at a verbal anticipatory-looking false-belief task. *British Journal of Developmental Psychology*, 30, 14–29. <http://doi.org/10.1111/j.2044-835X.2011.02070.x>
- Krupenye, C., Kano, F., Hirata, S., Call, J., & Tomasello, M. Great apes anticipate that other individuals will act according to false beliefs. *Science*, 354(6308), 110-114 (2016).

- Low, J. (2010). Preschoolers' implicit and explicit false-belief understanding: relations with complex syntactical mastery. *Child Development*, 81(2), 597–615.
<http://doi.org/10.1111/j.1467-8624.2009.01418.x>
- Meristo, M., Morgan, G., Geraci, A., Iozzi, L., Hjelmquist, E., Surian, L., & Siegal, M. (2012). Belief attribution in deaf and hearing infants. *Developmental Science*, 1–9.
<http://doi.org/10.1111/j.1467-7687.2012.01155.x>
- Milligan, K., Astington, J. W., & Dack, L. A. (2007). Language and theory of mind: meta-analysis of the relation between language ability and false-belief understanding. *Child Development*, 78(2), 622–46.
- Mischel, W., Shoda, Y., & Peake, P. K. (1988). The Nature of Adolescent Competencies Predicted by Preschool Delay of Gratification, 54(4), 687–696.
- Nathans, L. L., Oswald, F. L., & Nimon, K. (2012). Interpreting Multiple Linear Regression: A Guidebook of Variable Importance. *Practical assessment, Research & Evaluation* 17(9).
- Newson, R. S., & Kemps, E. B. (2005). The Journals of Gerontology Series B: Psychological Sciences and Social Sciences, 60(3), P113-P120.
- Oakes, L. M. (2012). Advances in eye tracking in infancy research. *Infancy*, 17(1), 1-8.
<http://doi.org/10.1111/j.1532-7078.2011.00101.x>
- Rakoczy, H. (2010). Executive function and the development of belief-desire psychology. *Developmental Science*, 13(4), 648–61. <http://doi.org/10.1111/j.1467-7687.2009.00922.x>
- Rodriguez, M. L., Mischel, W., & Shoda, Y. (1989). Cognitive Person Variables in the Delay of Gratification of Older Children at Risk, 57(2), 358–367.
- Seibold, D. R., & McPhee, R. D. (1979). *Human Communication Research*, 5(4), 355-365.
- Senju, A., Southgate, V., Snape, C., Leonard, M., & Csibra, G. (2011). Do 18-month-olds really attribute mental states to others? A critical test. *Psychological Science*, 22(7), 878–80.
<http://doi.org/10.1177/0956797611411584>
- Senju, A., Southgate, V., White, S., & Frith, U. (2009). Mindblind eyes: an absence of spontaneous theory of mind in Asperger syndrome. *Science (New York, N.Y.)*, 325(5942), 883–5.
<http://doi.org/10.1126/science.1176170>
- Shoda, Y., Mischel, W., & Peake, P. K. (1990). Predicting Adolescent Cognitive and Self-Regulatory Competencies From Preschool Delay of Gratification : Identifying Diagnostic Conditions, 26(6), 978–986.
- Southgate, V., Senju, A., & Csibra, G. (2007). Action anticipation through attribution of false belief by 2-year-olds. *Psychological Science*, 18(7), 587–92.
<http://doi.org/10.1111/j.1467-9280.2007.01944.x>
- Steinbeis, N., Bernhardt, B. C., & Singer, T. (2012). *Neuron*, 73(5), 1040-1051.
- Steinbeis, N., Bernhardt, B. C., & Singer, T. (2014). Social cognitive and affective neuroscience,

nsu057.

- Surian, L., & Geraci, A. (2012). Where will the triangle look for it? Attributing false beliefs to a geometric shape at 17 months, *4*, 30–44. <http://doi.org/10.1111/j.2044-835X.2011.02046.x>
- Thoermer, C., Sodian, B., Vuori, M., Perst, H., & Kristen, S. (2012). Continuity from an implicit to an explicit understanding of false belief from infancy to preschool age. *British Journal of Developmental Psychology*, *30*(1), 172-187.
- Van den Bos, W., Rodriguez, C. A., Schweitzer, J. B., & McClure, S. M. (2015). Proceedings of the National Academy of Sciences, *112*(29), E3765-E3774.
- Van den Bos, W., Cohen, M. X., Kahnt, T., & Crone, E. A. (2012). *Cerebral Cortex*, *22*(6), 1247-1255.
- Wellman, H. M., Cross, D., & Watson, J. (2011). Meta-analysis of theory-of-mind development: the truth about false belief. *Child Development*, *72*(3), 655–84. Retrieved from <http://www.ncbi.nlm.nih.gov/pubmed/11405571>
- Wellman, H. M., & Liu, D. (2004). Scaling of Theory-of-Mind Tasks. *Child Development*, *75*(2), 523–541.
- Winkler AM, Ridgway GR, Webster MA, Smith SM, Nichols TE. Permutation inference for the general linear model. *NeuroImage*, 2014;92:381-397.

References from the manuscript:

9. Schurz, M., Radua, J., Aichhorn, M., Richlan, F. & Perner, J. Fractionating theory of mind: a meta-analysis of functional brain imaging studies. *Neurosci. Biobehav. Rev.* **42**, 9–34 (2014).
10. Carrington, S. J. & Bailey, A. J. Are there theory of mind regions in the brain? A review of the neuroimaging literature. *Hum. Brain Mapp.* **30**, 2313–2335 (2009).
11. Saxe, R. R., Whitfield-gabrieli, S., Scholz, J. & Pelphrey, K. A. Brain Regions for Perceiving and Reasoning About Other People in School-Aged Children. **80**, 1197–1209 (2009).
12. Gweon, H., Dodell-Feder, D., Bedny, M. & Saxe, R. Theory of Mind Performance in Children Correlates With Functional Specialization of a Brain Region for Thinking About Thoughts. *Child Dev.* **83**, 1853–68 (2012).
13. Kobayashi, C., Glover, G. H. & Temple, E. Cultural and linguistic effects on neural bases of ‘Theory of Mind’ in American and Japanese children. *Brain Res.* **1164**, 95–107 (2007).
14. Sommer, M. *et al.* Modulation of the cortical false belief network during development. *Brain Res.* **1354**, 123–131 (2010).
15. Sabbagh, M. a., Bowman, L. C., Evraire, L. E. & Ito, J. M. B. Neurodevelopmental correlates of theory of mind in preschool children. *Child Dev.* **80**, 1147–1162 (2009).
16. Liu, D., Sabbagh, M. a., Gehring, W. J. & Wellman, H. M. Neural correlates of children’s theory of mind development. *Child Dev.* **80**, 318–326 (2009).

17. Herbet, G. *et al.* Inferring a dual-stream model of mentalizing from associative white matter fibres disconnection. *Brain* **137**, 944–959 (2014).
18. Ameis, S. H. & Catani, M. Altered white matter connectivity as a neural substrate for social impairment in Autism Spectrum Disorder. *Cortex* **62**, 158–181 (2015).
30. Onishi, K. H. & Baillargeon, R. Do 15-month-old infants understand false beliefs? *Science* **308**, 255–8 (2005).
31. Southgate, V., Senju, A. & Csibra, G. Action anticipation through attribution of false belief by 2-year-olds. *Psychol. Sci.* **18**, 587–592 (2007).
65. Tournier, J. D., Calamante, F. & Connelly, A. MRtrix: Diffusion tractography in crossing fiber regions. *Int. J. Imaging Syst. Technol.* **22**, 53–66 (2012).
66. Tournier, J. D., Calamante, F., Gadian, D. G. & Connelly, A. Direct estimation of the fiber orientation density function from diffusion-weighted MRI data using spherical deconvolution. *Neuroimage* **23**, 1176–1185 (2004).
67. Corbetta M., Miezin F.M., Dobmeyer S., Shulman G.L., & Petersen S.E. Attentional modulation of neural processing of shape, color, and velocity in humans. *Science* **22**, 1556-1559 (1990).
68. Devlin J.T., et al. Is there an anatomical basis for category-specificity? Semantic memory studies in PET and fMRI. *Neuropsychologia* **40**, 54-75 (2002).
69. Friederici A.D., Brauer J., Lohmann G. Maturation of the Language Network: From Inter- to Intrahemispheric Connectivities. *PLoS ONE* **6**, e20726 (2011).

REVIEWERS' COMMENTS:

Reviewer #1 (Remarks to the Author):

My view that the manuscript makes an interesting contribution to the literature hasn't changed. The authors have addressed my more minor concerns regarding the relations between this literature and the prior literature.

I was not, however, convinced by their rationale for maintaining their non-standard coding of the false belief task. Although I agree that variance could represent fragile understandings in principle (for instance, with multiple tasks) I don't agree that the same reasoning applies to multiple related questions within the same task. Carry over effects or pragmatic pressure to provide a different answer to two very similar questions are all factors that could be contributing to variability and make the positive findings difficult to interpret. I do not see how anything included in the response mitigates this concern. Measures of construct reliability (e.g., alpha) do not address issues of construct validity and that is what this is about. The authors state that the findings are not substantially different with the more traditional scoring, and so I would recommend that they report these.

I certainly appreciate the authors attempt to reduce the causal language in the paper given their cross-sectional design. Their new title, however, remains problematic. The paper says nothing about "how" theory of mind develops -- only that age-related maturational changes are associated with performance. Given that the causal nature of the association is not established, then also not established are the causal mechanisms underlying that association. The terms "how" and "role" implies a that white matter maturation causes theory of mind development and that is not established.

Reviewer #2 (Remarks to the Author):

The authors have done a great job addressing my concerns. This paper will make a strong contribution to the literature.

Reviewer #3 (Remarks to the Author):

I thank the authors for their thorough and adequate response to my comments.

Reviewer #1 (Remarks to the Author):

I was not, however, convinced by their rationale for maintaining their non-standard coding of the false belief task. Although I agree that variance could represent fragile understandings in principle (for instance, with multiple tasks) I don't agree that the same reasoning applies to multiple related questions within the same task. Carry over effects or pragmatic pressure to provide a different answer to two very similar questions are all factors that could be contributing to variability and make the positive findings difficult to interpret. I do not see how anything included in the response mitigates this concern. Measures of construct reliability (e.g., alpha) do not address issues of construct validity and that is what this is about. The authors state that the findings are not substantially different with the more traditional scoring, and so I would recommend that they report these.

We acknowledge the Reviewer's concern about potential pragmatic pressure to provide a different answer to two very similar questions. We strongly feel, however, that it would be problematic only to select data from a single question, and that simply dropping the other assessed answers lacks justification. Moreover, we believe that the extremely high correlation between the items of the same task (average correlation: $r = .89$) strongly speaks against the influence of pragmatic pressure to give different answers to the questions. The high correlation between the items indicates that these assess the same construct, and therefore supports the validity of our measure. To reconcile the Reviewer's view with our concern about dropping 2/3 of the assessed data without sufficient justification, we now discuss the Reviewer's concern in the paper and also report the results for the alternative scoring suggested by the Reviewer.

Methods, p. 16:

The three questions in each of the false belief tasks could have led to carry-over effects or pragmatic pressure to give different answers to consecutive very similar questions. To exclude the possibility that such effects might have influenced our results, we replicated our analyses with a false belief score that only included the first question of each of the two false belief tasks. This scoring yielded very similar results (see Supplementary Tables 2 and 3).

I certainly appreciate the authors attempt to reduce the causal language in the paper given their cross-sectional design. Their new title, however, remains problematic. The paper says nothing about "how" theory of mind develops -- only that age-related maturational changes are associated with performance. Given that the causal nature of the association is not established, then also not established are the causal mechanisms underlying that association. The terms "how" and "role" implies a that white matter maturation causes theory of mind development and that is not established.

We have now changed the title to “White matter maturation is associated with the emergence of Theory of Mind in early childhood”.